# ToolLLM: Facilitating Large Language Models to Master 16000+ Real-world APIs

**Yujia Qin[1*], Shihao Liang[1*], Yining Ye[1], Kunlun Zhu[1], Lan Yan[1], Yaxi Lu[1], Yankai Lin[3†],
Xin Cong[1], Xiangru Tang[4], Bill Qian[4], Sihan Zhao[1], Lauren Hong[1], Runchu Tian[1],
Ruobing Xie[5], Jie Zhou[5], Mark Gerstein[4], Dahai Li[2,6], Zhiyuan Liu[1†], Maosong Sun[1†]**

[1]Tsinghua University [2]ModelBest Inc. [3]Renmin University of China
[4]Yale University [5]WeChat AI, Tencent Inc. [6]Zhihu Inc.
yujiaqin16@gmail.com

## Abstract

Despite the advancements of open-source large language models (LLMs), e.g., LLaMA, they remain significantly limited in tool-use capabilities, i.e., using external tools (APIs) to fulfill human instructions. The reason is that current instruction tuning largely focuses on basic language tasks but ignores the tool-use domain. This is in contrast to the excellent tool-use capabilities of state-of-the-art (SOTA) closed-source LLMs, e.g., ChatGPT. To bridge this gap, we introduce ToolLLM, a general tool-use framework encompassing data construction, model training, and evaluation. We first present ToolBench, an instruction-tuning dataset for tool use, which is constructed automatically using ChatGPT. Specifically, the construction can be divided into three stages: (i) API collection: we collect $16,464$ real-world RESTful APIs spanning $49$ categories from RapidAPI Hub; (ii) instruction generation: we prompt ChatGPT to generate diverse instructions involving these APIs, covering both single-tool and multi-tool scenarios; (iii) solution path annotation: we use ChatGPT to search for a valid solution path (chain of API calls) for each instruction. To enhance the reasoning capabilities of LLMs, we develop a novel depth-first search-based decision tree algorithm. It enables LLMs to evaluate multiple reasoning traces and expand the search space. Moreover, to evaluate the tool-use capabilities of LLMs, we develop an automatic evaluator: ToolEval. Based on ToolBench, we fine-tune LLaMA to obtain an LLM ToolLLaMA, and equip it with a neural API retriever to recommend appropriate APIs for each instruction. Experiments show that ToolLLaMA demonstrates a remarkable ability to execute complex instructions and generalize to unseen APIs, and exhibits comparable performance to ChatGPT. Our ToolLLaMA also demonstrates strong zero-shot generalization ability in an out-of-distribution tool-use dataset: APIBench. The codes, trained models, and demo are publicly available at https://github.com/OpenBMB/ToolBench.

## 1 Introduction

Tool learning (Qin et al., 2023b) aims to unleash the power of large language models (LLMs) to effectively interact with various tools (APIs) to accomplish complex tasks. By integrating LLMs with APIs, we can greatly expand their utility and empower them to serve as efficient intermediaries between users and the vast ecosystem of applications. Although open-source LLMs, e.g., LLaMA (Touvron et al., 2023a), have achieved versatile capabilities through instruction tuning (Taori et al., 2023; Chiang et al., 2023), they still lack the sophistication in performing higher-level tasks, such as appropriately interacting with tools (APIs) to fulfill complex human instruction. This deficiency is because current instruction tuning largely focuses on basic language tasks, with a relative neglect of the tool-use domain. On the other hand, current state-of-the-art (SOTA) LLMs (e.g., ChatGPT (OpenAI,

---

* Indicates equal contribution.
† Corresponding author.

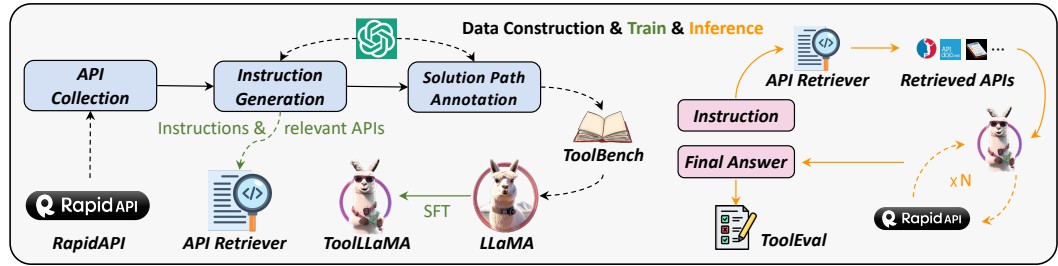

Figure 1: Three phases of constructing ToolBench and how we train our API retriever and ToolLLaMA. During inference of an instruction, the API retriever recommends relevant APIs to ToolLLaMA, which performs multiple rounds of API calls to derive the final answer. The whole reasoning process is evaluated by ToolEval.

2022) and GPT-4 (OpenAI, 2023)), which have demonstrated impressive competencies in utilizing tools (Bubeck et al., 2023), are closed-source with their inner mechanisms opaque. This limits the democratization of AI technologies and the scope of community-driven innovation and development. In this regard, we deem it urgent to *empower open-source LLMs to skillfully master diverse APIs.*

Although prior works have explored building instruction tuning data for tool use (Li et al., 2023a; Patil et al., 2023; Tang et al., 2023; Xu et al., 2023b), they fail to fully stimulate the tool-use capabilities within LLMs and have inherent limitations: (1) **limited APIs**: they either fail to involve real-world APIs (e.g., RESTAPI) (Patil et al., 2023; Tang et al., 2023) or consider only a small scope of APIs with poor diversity (Patil et al., 2023; Xu et al., 2023b; Li et al., 2023a);

(2) **constrained scenario**: existing works are confined to instructions that only involve one single tool. In contrast, real-world scenarios may require that multiple tools are interleaved together for multi-round tool execution to solve a complex task. Besides, they often assume that users manually specify the ideal API set for a given instruction in advance, which is infeasible with a large collection of real-world APIs; (3) **inferior planning and reasoning**: existing works adopted either CoT (Wei et al., 2023) or ReACT (Yao et al., 2022) for model reasoning, which cannot fully elicit the capabilities stored in LLMs and thus fail to handle complex instructions. In addition, some works do not even execute APIs to obtain real responses (Patil et al., 2023; Tang et al., 2023), which serve as important information for subsequent model planning.

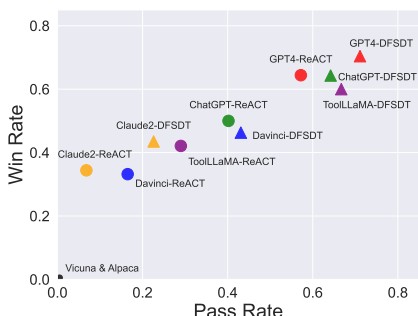

Figure 2: Pass rate (↑) and win rate (↑) of different methods in tool-use evaluation. For win rate, we compare each method with ChatGPT-ReACT. DFSDT is our improved reasoning strategy over ReACT. ToolLLaMA surpasses Text-Davinci-003, Claude-2, and almost performs on par with ChatGPT.

To facilitate tool-use capabilities within open-source LLMs, we introduce **ToolLLM**, a general tool-use framework including data construction, model training, and evaluation. As illustrated in Figure 1, we collect a high-quality instruction-tuning dataset **ToolBench**. It is constructed automatically using ChatGPT (*gpt-3.5-turbo-16k*), which has been upgraded with function call (link) capabilities. The comparison between ToolBench and prior works is listed in Table 1. Specifically, the construction of ToolBench entails three phases:

- **API Collection**: we gather **16,464** representational state transfer (REST) APIs from RapidAPI (link), a platform that hosts massive real-world APIs provided by developers. These APIs span **49** diverse categories such as social media, e-commerce, and weather. For each API, we crawl detailed API documents from RapidAPI, including the functionality descriptions, required parameters, code snippets for API calls, etc. By comprehending these documents to learn to execute APIs, LLMs can generalize to new APIs unseen during training;

- **Instruction Generation**: we first sample APIs from the whole set and then prompt ChatGPT to generate diverse instructions for these APIs. To cover practical scenarios, we curate instructions

| Resource | ToolBench (this work) | APIBench (Patil et al., 2023) | API-Bank (Li et al., 2023a) | ToolAlpaca (Tang et al., 2023) | ToolBench (Xu et al., 2023b) |
|---|---|---|---|---|---|
| Real-world API? | ✓ | ✗ | ✓ | ✗ | ✓ |
| Real API Call&Response? | ✓ | ✗ | ✓ | ✗ | ✓ |
| Multi-tool Scenario? | ✓ | ✗ | ✗ | ✗ | ✗ |
| API Retrieval? | ✓ | ✓ | ✗ | ✗ | ✓ |
| Multi-step Reasoning? | ✓ | ✗ | ✓ | ✓ | ✓ |
| Number of tools | **3451** | 3 | 53 | 400 | 8 |
| Number of APIs | **16464** | 1645 | 53 | 400 | 232 |
| Number of Instances | **126486** | 17002 | 274 | 3938 | 2746 |
| Number of Real API Calls | **469585** | 0 | 568 | 0 | 3926 |
| Avg. Reasoning Traces | 4.0 | 1.0 | 2.1 | 1.0 | **5.9** |

Table 1: A comparison of our ToolBench to notable instruction tuning dataset for tool learning.

that involve both **single-tool** and **multi-tool** scenarios. This ensures that our model learns not only how to interact with individual tools but also how to combine them to accomplish complex tasks;

- **Solution Path Annotation**: each solution path may contain multiple rounds of model reasoning and real-time API calls to derive the final response. However, even the most sophisticated LLM, i.e., GPT-4, achieves a low pass rate for complex human instructions, making annotation inefficient. To this end, we develop a novel **depth-first search-based decision tree** (DFSDT) to bolster the planning and reasoning ability of LLMs. Compared with conventional ReACT, DFSDT enables LLMs to evaluate a multitude of reasoning paths and make deliberate decisions to either retract steps or proceed along a promising path. In experiments, DFSDT significantly improves the annotation efficiency and successfully completes those complex instructions that cannot be fulfilled using ReACT.

To assess the tool-use capabilities of LLMs, we develop an automatic evaluator, **ToolEval**, backed up by ChatGPT. It comprises two key metrics: (1) *pass rate*, which measures LLM's ability to successfully execute an instruction within limited budgets, and (2) *win rate*, which compares the quality and usefulness of two solution paths. We demonstrate that ToolEval achieves a high correlation with human evaluation and provides a robust, scalable, and reliable assessment for machine tool use.

By fine-tuning LLaMA on ToolBench, we obtain **ToolLLaMA**. After evaluation based on our ToolEval, we derive the following findings:

- ToolLLaMA demonstrates a compelling capability to handle both single-tool and complex multi-tool instructions. As depicted in Figure 2, ToolLLaMA outperforms Text-Davinci-003 and Claude-2, achieves comparable performance to the "teacher model" ChatGPT, and is only slightly inferior to GPT4. Besides, ToolLLaMA exhibits **robust generalization to previously unseen APIs**, requiring only the API documentation to adapt to new APIs effectively. This flexibility allows users to incorporate novel APIs seamlessly, thus enhancing the model's practical utility.

- We show that our DFSDT serves as a general decision-making strategy to enhance the reasoning capabilities of LLMs. DFSDT broadens the search space by considering multiple reasoning traces and achieves significantly better performance than ReACT.

- We train a neural **API retriever**, which alleviates the need for manual selection from the large API pool in practice. As shown in Figure 1, given an instruction, the API retriever recommends a set of relevant APIs, which are sent to ToolLLaMA for multi-round decision making to derive the final answer. Despite sifting through a large pool of APIs, the retriever exhibits remarkable retrieval precision, returning APIs closely aligned with the ground truth.

- ToolLLaMA exhibits strong **generalization** performance on an **out-of-distribution** (OOD) dataset APIBench (Patil et al., 2023). Despite not training on any of the APIs or instructions on APIBench, ToolLLaMA performs on par with Gorilla, a pipeline specifically designed for APIBench.

## 2 DATASET CONSTRUCTION

We introduce the three-stage construction process of ToolBench: API collection (§ 2.1), instruction generation (§ 2.2), and solution path annotation (§ 2.3). All procedures are based on ChatGPT (*gpt-3.5-turbo-16k*), requiring minimal human supervision and can be easily extended to new APIs.

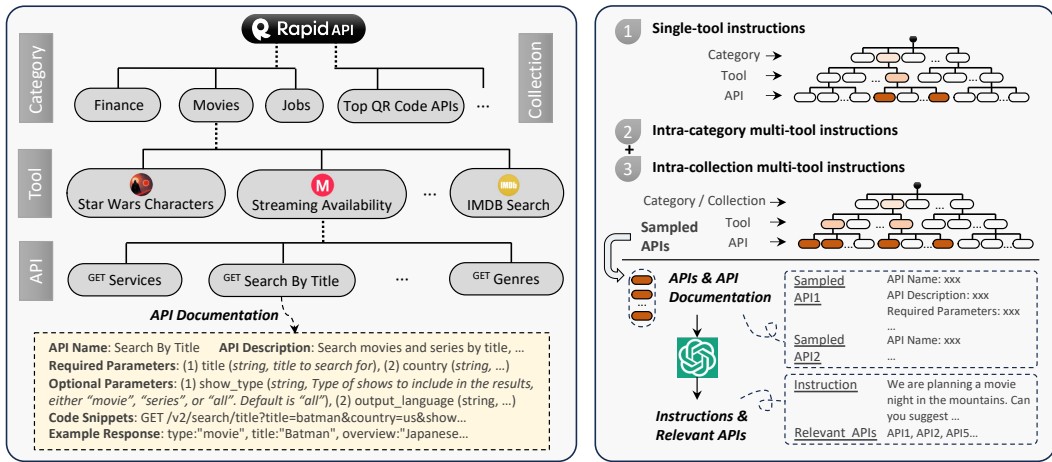

Figure 3: The hierarchy of RapidAPI (left) and the process of instruction generation (right).

## 2.1 API COLLECTION

We start by introducing RapidAPI and its hierarchy, followed by how we crawl and filter APIs.

**RapidAPI Hub** RapidAPI is a leading API marketplace that connects developers with thousands of real-world APIs, streamlining the process of integrating diverse services into applications. Developers can test and connect with various APIs by registering only a RapidAPI key. All APIs in RapidAPI can be classified into 49 *coarse-grained* **categories** (link), such as sports, finance, and weather. The categories associate an API with the most relevant topic. Additionally, the hub also provides $500+$ *fine-grained* categorization called **collections** (link), e.g., Chinese APIs and database APIs. APIs in the same collection share a common characteristic and often have similar functionalities or goals.

**Hierarchy of RapidAPI** As shown in Figure 3, each tool may be composed of multiple APIs. For each tool, we crawl the following information: the name and description of the tool, the URL of the host, and all the available APIs belonging to the tool; for each API, we record its name, description, HTTP method, required parameters, optional parameters, request body, executable code snippets for API call, and an example API call response. This rich and detailed metadata serves as a valuable resource for LLMs to understand and effectively use the APIs, even in a zero-shot manner.

**API Filtering** Initially, we gathered $10,853$ tools ($53,190$ APIs) from RapidAPI. However, the quality and reliability of these APIs can vary significantly. In particular, some APIs may not be well-maintained, such as returning 404 errors or other internal errors. To this end, we perform a rigorous filtering process (details in appendix A.1) to ensure that the ultimate tool set of ToolBench is reliable and functional. Finally, we only retain $3,451$ high-quality tools ($16,464$ APIs).

## 2.2 INSTRUCTION GENERATION

Different from prior works, we specifically focus on two crucial aspects for instruction generation: (1) **diversity**: to train LLMs to handle a wide range of API usage scenarios, thereby boosting their generalizability and robustness; and (2) **multi-tool usage**: to mirror real-world situations that often demand the interplay of multiple tools, improving the practical applicability and flexibility of LLMs. To this end, instead of brainstorming instructions from scratch and then searching for relevant APIs, we sample different combinations of APIs and craft various instructions that involve them.

**Generating Instructions for APIs** Define the total API set as $\mathbb{S}_{\text{API}}$, at each time, we sample a few APIs: $\mathbb{S}_N^{\text{sub}} = \{\text{API}_1, \cdots, \text{API}_N\}$ from $\mathbb{S}_{\text{API}}$. We prompt ChatGPT to understand the functionalities of these APIs and then generate (1) possible instructions ($\text{Inst}_*$) that involve APIs in $\mathbb{S}_N^{\text{sub}}$, and (2) relevant APIs ($\mathbb{S}_*^{\text{rel}} \subset \mathbb{S}_N^{\text{sub}}$) for each instruction ($\text{Inst}_*$), i.e., $\{[\mathbb{S}_1^{\text{rel}}, \text{Inst}_1], \cdots, [\mathbb{S}_{N'}^{\text{rel}}, \text{Inst}_{N'}]\}$, where $N'$ denotes the number of generated instances. These (instruction, relevant API) pairs will be used for

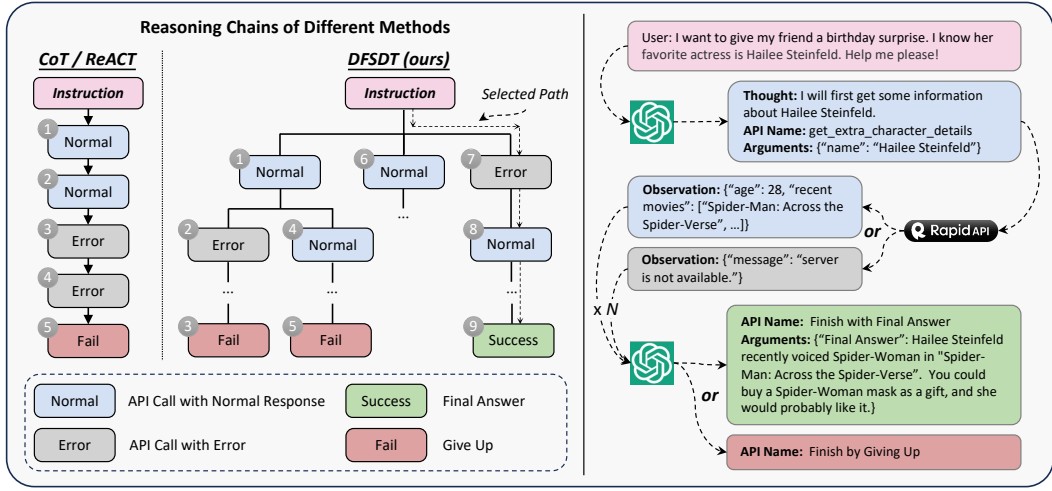

Figure 4: A comparison of our DFSDT and conventional CoT or ReACT during model reasoning (left). We show part of the solution path annotation process using ChatGPT (right).

training the API retriever in § 3.1. We use different sampling strategies (introduced later) to cover all APIs and most of their combinations, thus ensuring the diversity of our instructions.

The prompt for ChatGPT is composed of (1) a general description of the intended instruction generation task, (2) comprehensive documentation of each API in $\mathbb{S}_N^{sub}$, which helps ChatGPT understand their functionality and interplay, and (3) three in-context seed examples $\{seed_1, seed_2, seed_3\}$. Each seed example is an ideal instruction generation written by human experts. These seed examples are leveraged to better regulate ChatGPT's behavior through in-context learning. In total, we wrote 12 / 36 diverse seed examples ($\mathbb{S}_{seed}$) for the single-tool / multi-tool setting, and randomly sampled three examples at each time. Detailed prompts for instruction generation are described in appendix A.7. Overall, the generation process can be formulated as follows:

$$\underset{\{API_1, \cdots, API_N\} \in \mathbb{S}_{API}, \{seed_1, \cdots, seed_3\} \in \mathbb{S}_{seed}}{\text{ChatGPT}} (\{[\mathbb{S}_1^{rel}, Inst_1], \cdots, [\mathbb{S}_{N'}^{rel}, Inst_{N'}]\} | API_1, \cdots, API_N, seed_1, \cdots, seed_3).$$

**Sampling Strategies for Different Scenarios** As shown in Figure 3, for the **single-tool instructions (I1)**, we iterate over each tool and generate instructions for its APIs. However, for the **multi-tool setting**, since the interconnections among different tools in RapidAPI are sparse, random sampling tool combinations from the whole tool set often leads to a series of irrelevant tools that cannot be covered by a single instruction in a natural way. To address the sparsity issue, we leverage the RapidAPI hierarchy information. Since tools belonging to the same RapidAPI *category* or *collection* are generally related to each other in the functionality and goals, we randomly select 2-5 tools from the same category / collection and sample at most 3 APIs from each tool to generate the instructions. We denote the generated instructions as **intra-category multi-tool instructions (I2)** and **intra-collection multi-tool instructions (I3)**, respectively. Through rigorous human evaluation, we find that instructions generated in this way already have a high diversity that covers various practical scenarios. We also provide visualization for instructions using Atlas (link) to support our claim.

After generating the initial set of instructions, we further filter those with the hallucinated relevant APIs by assessing whether they exist in $\mathbb{S}_N^{sub}$. Finally, we collect nearly 200k qualified (instruction, relevant API) pairs, including 87413, 84815, and 25251 instances for I1, I2, and I3, respectively.

## 2.3 SOLUTION PATH ANNOTATION

As shown in Figure 4, given an instruction $Inst_*$, we prompt ChatGPT to search for a valid action sequence: $\{a_1, \cdots, a_N\}$. Such a multi-step decision-making process is cast as a multi-round conversation for ChatGPT. At each round $t$, the model generates an action $a_t$ based on previous interactions, i.e., $\text{ChatGPT}(a_t | \{a_1, r_1, \cdots, a_{t-1}, r_{t-1}\}, Inst_*)$, where $r_*$ denotes the real API response. For each

$a_t$, ChatGPT should specify its "thought", which API to use, and the specific parameters for this API, i.e., $a_t$ has the following format: "`Thought: ⋯, API Name: ⋯, Parameters: ⋯`".

To leverage the **function call** feature of ChatGPT, we treat each API as a special function and feed its API documentation into ChatGPT's function field. In this way, the model understands how to call the API. For each instruction Inst$_*$, we feed all the sampled APIs $\mathbb{S}_N^{sub}$ to ChatGPT's as available functions. To let ChatGPT finish an action sequence, we define two additional functions, i.e., "Finish with Final Answer" and "Finish by Giving Up". The former function has a parameter that corresponds to a detailed final answer to the original instruction; while the latter function is designed for cases where the provided APIs cannot complete the original instruction after multiple API call attempts.

**Depth First Search-based Decision Tree** In our pilot studies, we find that CoT (Wei et al., 2023) or ReACT (Yao et al., 2022) has inherent limitations: (1) **error propagation**: a mistaken action may propagate the errors further and cause the model to be trapped in a faulty loop, such as continually calling an API in a wrong way or hallucinating APIs; (2) **limited exploration**: CoT or ReACT only explores one possible direction, leading to limited exploration of the whole action space. Hence even GPT-4 often fails to find a valid solution path, making annotation difficult.

To this end, we propose to construct a decision tree to expand the search space and increase the possibility of finding a valid path. As depicted in Figure 4, our DFSDT allows the model to assess different reasoning paths and choose to either (1) proceed along a promising path or (2) abandon an existing node by calling the "Finish by Giving Up" function and expand a new node. During node expansion, to diversify the child nodes and expand the search space, we prompt ChatGPT with the information of the previously generated nodes and explicitly encourage the model to generate a distinct node. For the searching process, we prefer depth-first search (DFS) instead of breadth-first search (BFS) because the annotation can be finished as long as one valid path is found. Using BFS will cost excessive OpenAI API calls. More details are described in appendix A.8. We perform DFSDT for all the generated instructions and only retain those passed solution paths. Ultimately, we generate $126,486$ (instruction, solution path) pairs, which are used to train ToolLLaMA in § 3.2.

## 3 EXPERIMENTS

In this section, we investigate the performance of ToolLLM framework. We first introduce the evaluation metric and evaluate the efficacy of API retriever and DFSDT in § 3.1. Then we present the main experiments in § 3.2, followed by a generalization experiment in § 3.3.

### 3.1 PRELIMINARY EXPERIMENTS

**ToolEval** Considering the API's temporal variability on RapidAPI and the infinite potential solution paths for an instruction, it is infeasible to annotate a fixed ground-truth solution path for each test instruction. Considering that human evaluation can be time-consuming, we follow AlpacaEval (Li et al., 2023b) to develop an efficient evaluator **ToolEval** based on ChatGPT, which incorporates two evaluation metrics (details in appendix A.5): (1) **Pass Rate**: it calculates the proportion of successfully completing an instruction within limited budgets. The metric measures the executability of instructions for an LLM and can be seen as a basic requirement for ideal tool use; and (2) **Win Rate**: we provide an instruction and two solution paths to ChatGPT evaluator and obtain its preference (i.e., which one is better). We pre-define a set of criteria for both metrics and these criteria are organized as prompts for our ChatGPT evaluator. We evaluate multiple times based on ChatGPT to improve the reliability. Then we calculate the average results from the evaluator.

Through rigorous testing (details in appendix A.5), we find that ToolEval demonstrates a high agreement of $87.1\%$ in pass rate and $80.3\%$ in win rate with human annotators. This shows that ToolEval can reflect and represent human evaluation to a large extent.

**Efficacy of API Retriever** The API retriever aims to retrieve relevant APIs to an instruction. We employ Sentence-BERT (Reimers & Gurevych, 2019) to train a dense retriever based on BERT-BASE (Devlin et al., 2019). The API retriever encodes the instruction and API document into two embeddings, and calculates their relevance with embedding similarity. For training, we regard the relevant APIs of each instruction generated in § 2.2 as positive examples and sample a few other APIs as negative examples for contrastive learning. For baselines, we choose BM25 (Robertson et al.,

| Method | I1 NDCG | | I2 NDCG | | I3 NDCG | | Average NDCG | |
|--------|------|------|------|------|------|------|------|------|
|        | @1   | @5   | @1   | @5   | @1   | @5   | @1   | @5   |
| BM25   | 18.4 | 19.7 | 12.0 | 11.0 | 25.2 | 20.4 | 18.5 | 17.0 |
| Ada    | 57.5 | 58.8 | 36.8 | 30.7 | 54.6 | 46.8 | 49.6 | 45.4 |
| Ours   | 84.2 | 89.7 | 68.2 | 77.9 | 81.7 | 87.1 | 78.0 | 84.9 |

Table 2: Our API retriever v.s. two baselines for three types of instructions (I1, I2, I3). We report NDCG@1 and NDCG@5.

| Method   | I1   | I2   | I3   | Average |
|----------|------|------|------|---------|
| ReACT    | 37.8 | 40.6 | 27.6 | 35.3    |
| ReACT@N  | 49.4 | 49.4 | 34.6 | 44.5    |
| DFSDT    | 58.0 | 70.6 | 62.8 | 63.8    |

Table 3: Pass rate of different reasoning strategies for three types of instructions (I1, I2, I3) based on ChatGPT.

2009) and OpenAI's *text-embedding-ada-002* (link). We evaluate the retrieval performance using NDCG (Järvelin & Kekäläinen, 2002). We train and evaluate our model on single-tool instructions (I1), intra-category multi-tool instructions (I2), and intra-collection multi-tool instructions (I3).

As shown in Table 2, our API retriever consistently outperforms baselines across all settings, indicating its feasibility in real-world scenarios with massive APIs. Also, the NDCG score of I1 is generally higher than I2 and I3, which means single-tool instruction retrieval is simpler than multi-tool setting.

**Superiority of DFSDT over ReACT**   Before solution path annotation, we validate the efficacy of DFSDT. Based on ChatGPT, we compare DFSDT and ReACT using the pass rate metric. Since DFSDT consumes more OpenAI API calls than ReACT, for a fairer comparison, we also establish a "ReACT@N" baseline, which conducts multiple times of ReACT until the total costs reach the same level of DFSDT. Once a valid solution is found by ReACT@N, we deem it a pass.

From Table 3, it can be observed that DFSDT significantly outperforms the two baselines in all scenarios. Since we only retain those passed annotations as the training data, given the same budgets, using DFSDT could annotate more instructions. This makes DFSDT a more efficient way that saves the total annotation cost. We also find that the performance improvement of DFSDT is more evident for harder instructions (i.e., I2 and I3) than those simpler instructions (I1). This means that by expanding the search space, DFSDT can better solve those difficult, complex instructions that are unanswerable by the vanilla ReACT no matter how many times it is performed.

## 3.2 MAIN EXPERIMENTS

**ToolLLaMA**   We fine-tune LLaMA-2 7B model (Touvron et al., 2023b) using the instruction-solution pairs. The original LLaMA-2 model has a sequence length of $4096$, which is not enough under our setting since the API response can be very long. To this end, we use positional interpolation (Chen et al., 2023) to extend the context length to $8192$ (training details in appendix A.3).

**Settings**   Ideally, by scaling the number and diversity of instructions and unique tools in the training data, ToolLLaMA is expected to generalize to new instructions and APIs unseen during training. This is meaningful since users can define customized APIs and expect ToolLLaMA to adapt according to the documentation. To this end, we strive to evaluate the **generalization ability** of ToolLLaMA at three levels: (1) **Inst.**: **unseen instructions** for the same set of tools in the training data, (2) **Tool**: **unseen tools** that belong to the **same (seen) category** of the tools in the training data, and (3) **Cat.**: **unseen tools** that belong to a **different (unseen) category** of tools in the training data.

We perform experiments on three scenarios: single-tool instructions (I1), intra-category multi-tool instructions (I2), and intra-collection multi-tool instructions (I3). For I1, we conduct the evaluation for the aforementioned three levels (I1-Inst., I1-Tool, and I1-Cat.); for I2, since the training instructions already involve different tools of the same category, we only perform level 1 and level 3 for the generalization evaluation (I2-Inst. and I2-Cat.); similarly, we only perform level 1 generalization for I3 (I3-Inst.) since it already covers instructions that involve various combinations of tools from different categories (the tools in a RapidAPI collection may come from different RapidAPI categories). For each test instruction, we feed the ground-truth (oracle) APIs $\mathbb{S}_N^{sub}$ to each model. This simulates the scenario where the user specifies the API set they prefer.

**Baselines**   We choose two LLaMA variants that have been fine-tuned for general-purpose dialogue, i.e., Vicuna (Chiang et al., 2023) and Alpaca (Taori et al., 2023). We also choose the "teacher model" ChatGPT, Text-Davinci-003, GPT-4, and Claude-2 as baselines, and apply both DFSDT and ReACT to them. When calculating the win rate, each model is compared with ChatGPT-ReACT.

| Model | Method | I1-Inst. | | I1-Tool | | I1-Cat. | | I2-Inst. | | I2-Cat. | | I3-Inst. | | Average | |
|---|---|---|---|---|---|---|---|---|---|---|---|---|---|---|---|
| | | Pass | Win | Pass | Win | Pass | Win | Pass | Win | Pass | Win | Pass | Win | Pass | Win |
| ChatGPT | ReACT | 41.5 | - | 44.0 | - | 44.5 | - | 42.5 | - | 46.5 | - | 22.0 | - | 40.2 | - |
| | DFSDT | 54.5 | 60.5 | 65.0 | 62.0 | 60.5 | 57.3 | 75.0 | 72.0 | 71.5 | 64.8 | 62.0 | 69.0 | 64.8 | 64.3 |
| Claude-2 | ReACT | 5.5 | 31.0 | 3.5 | 27.8 | 5.5 | 33.8 | 6.0 | 35.0 | 6.0 | 31.5 | 14.0 | 47.5 | 6.8 | 34.4 |
| | DFSDT | 20.5 | 38.0 | 31.0 | 44.3 | 18.5 | 43.3 | 17.0 | 36.8 | 20.5 | 33.5 | 28.0 | 65.0 | 22.6 | 43.5 |
| Text-Davinci-003 | ReACT | 12.0 | 28.5 | 20.0 | 35.3 | 20.0 | 31.0 | 8.5 | 29.8 | 14.5 | 29.8 | 24.0 | 45.0 | 16.5 | 33.2 |
| | DFSDT | 43.5 | 40.3 | 44.0 | 43.8 | 46.0 | 46.8 | 37.0 | 40.5 | 42.0 | 43.3 | 46.0 | 63.0 | 43.1 | 46.3 |
| GPT4 | ReACT | 53.5 | 60.0 | 50.0 | 58.8 | 53.5 | 63.5 | 67.0 | 65.8 | 67.0 | 65.8 | 47.0 | 78.0 | 57.2 | 64.4 |
| | DFSDT | 60.0 | 67.5 | 71.5 | 67.8 | 67.0 | 66.5 | 79.5 | 73.3 | 77.5 | 63.3 | 71.0 | 84.0 | 71.1 | 70.4 |
| Vicuna | ReACT & DFSDT | 0.0 | 0.0 | 0.0 | 0.0 | 0.0 | 0.0 | 0.0 | 0.0 | 0.0 | 0.0 | 0.0 | 0.0 | 0.0 | 0.0 |
| Alpaca | ReACT & DFSDT | 0.0 | 0.0 | 0.0 | 0.0 | 0.0 | 0.0 | 0.0 | 0.0 | 0.0 | 0.0 | 0.0 | 0.0 | 0.0 | 0.0 |
| | ReACT | 25.0 | 45.0 | 29.0 | 42.0 | 33.0 | 47.5 | 30.5 | 50.8 | 31.5 | 41.8 | 25.0 | 55.0 | 29.0 | 47.0 |
| ToolLLaMA | DFSDT | 57.0 | 55.0 | 61.0 | 55.3 | 62.0 | 54.5 | 77.0 | 68.5 | 77.0 | 58.0 | 66.0 | 69.0 | 66.7 | 60.0 |
| | DFSDT-Retriever | 64.0 | 62.3 | 64.0 | 59.0 | 60.5 | 55.0 | 81.5 | 68.5 | 68.5 | 60.8 | 65.0 | 73.0 | 67.3 | 63.1 |

Table 4: Main experiments of ToolBench. Win rate is calculated by comparing each model with ChatGPT-ReACT. A win rate higher than $50\%$ means the model performs better than ChatGPT-ReACT. Apart from ToolLLaMA-DFSDT-Retriever, all methods use the oracle API retriever (i.e., ground truth API).

**Main Results**     The results are placed in Table 4, from which we derive that:

1. Both Vicuna and Alpaca fail to pass any instruction (pass rate & win rate = 0), which means their instruction-following abilities do not cover the tool-use domain. This underscores **the deficiency of current instruction tuning attempts**, which largely focus on language skills;
2. For all LLMs, using DFSDT significantly outperforms ReACT in both pass rate and win rate. Notably, ChatGPT +DFSDT surpasses GPT-4+ReACT in pass rate and performs comparably in win rate. This underscores **the superiority of DFSDT over ReACT** in decision-making;
3. When using DFSDT, ToolLLaMA performs much better than Text-Dainci-003 and Claude-2, and achieves a result almost on par with ChatGPT (the teacher model). In general, despite generalizing to unseen instructions and tools, ToolLLaMA +DFSDT demonstrates **competitive generalization performance** in all scenarios, achieving a pass rate second to GPT4+DFSDT.

Overall, these results demonstrate that ToolBench can sufficiently elicit the tool-use capabilities within LLMs and empower them to skillfully master even unseen APIs for various instructions.

**Integrating API Retriever with ToolLLaMA**     In real-world scenarios, asking users to manually recommend APIs from a large pool may not be practical. To emulate this practical setting, we feed the top 5 APIs (instead of the ground truth APIs $\mathbb{S}_N^{sub}$) recommended by our API retriever to ToolLLaMA. As shown in Table 4, using retrieved APIs even improves the performance compared to the ground truth API set. This is because many APIs in the ground truth API set can be replaced by other similar APIs with better functionalities, which our API retriever can successfully identify. In other words, **our retriever expands the search space of relevant APIs and finds more appropriate ones for the current instruction**. It demonstrates the excellent ability of our API retriever to retrieve relevant APIs, especially considering the vast pool $(16,000+)$ of APIs from which our API retriever selects.

### 3.3    OUT-OF-DISTRIBUTION (OOD) GENERALIZATION TO APIBENCH (PATIL ET AL., 2023)

**Settings**     We further extend ToolLLaMA to an OOD dataset APIBench to validate its generalization ability. We equip ToolLLaMA with two retrievers: our trained API retriever and the oracle retriever. We evaluate three domains of APIBench, i.e., TorchHub, TensorHub, and HuggingFace. We compare ToolLLaMA with Gorilla, a LLaMA-7B model fine-tuned using the training data of APIBench. Following the original paper, we adopt two settings for Gorilla: zero-shot setting (ZS) and retrieval-aware setting (RS). The latter means (RS) the retrieved APIs are sent to the model as part of the prompts; while the former (ZS) does not incorporate the APIs in the prompts when training the model. We adopt the official evaluation metric and report the AST accuracy and the hallucination rates.

**Results**     The results are shown in Table 5. In general, ToolLLaMA achieves **remarkable OOD generalization performance** on all three datasets, despite being trained on a completely different API domain and instruction domain. Specifically, ToolLLaMA+our API retriever outperforms Gorilla+BM25 from both training settings (ZS / RS) in terms of AST accuracy on HuggingFace and TorchHub. With the same oracle retriever, ToolLLaMA is consistently superior when compared to Gorilla-ZS. It should be noted that Gorilla model cannot be generalized to our ToolBench dataset due to our more complex settings, such as the multi-tool use and multi-step reasoning.

| Method | HuggingFace | | TorchHub | | TensorHub | |
|---|---|---|---|---|---|---|
| | Hallu. ($\downarrow$) | AST ($\uparrow$) | Hallu. ($\downarrow$) | AST ($\uparrow$) | Hallu. ($\downarrow$) | AST ($\uparrow$) |
| ToolLLaMA + Our Retriever | 10.60 | **16.77** | 15.70 | **51.16** | 6.48 | 40.59 |
| Gorilla-ZS + BM25 | 46.90 | 10.51 | 17.20 | 44.62 | 20.58 | 34.31 |
| Gorilla-RS + BM25 | **6.42** | 15.71 | **5.91** | 50.00 | **2.77** | **41.90** |
| ToolLLaMA + Oracle | 8.66 | 88.80 | 14.12 | 85.88 | 7.44 | 88.62 |
| Gorilla-ZS + Oracle | 52.88 | 44.36 | 39.25 | 59.14 | 12.99 | 83.21 |
| Gorilla-RS + Oracle | **6.97** | **89.27** | **6.99** | **93.01** | **2.04** | **94.16** |

Table 5: OOD generalization experiments on APIBench. For the Gorilla entries, ZS / RS means that Gorilla was trained in a zero-shot / retrieval-aware setting on APIBench. We report hallucination rate and AST accuracy.

## 4 RELATED WORK

**Tool Learning** Recent studies have shed light on the burgeoning capabilities of LLMs in mastering tools and making decisions within complex environments (Nakano et al., 2021; Qin et al., 2023a; Shen et al., 2023; Wu et al., 2023; Schick et al., 2023; Hao et al., 2023; Qian et al., 2023; Song et al., 2023; Zhuang et al., 2023; Gao et al., 2023). Gaining access to external tools endows LLMs with real-time factual knowledge (Yang et al., 2023), multimodal functionalities (Gupta & Kembhavi, 2023), and specialized skills in vertical domains (Jin et al., 2023). However, open-source LLMs still lag far behind SOTA LLMs in tool use, and how tool-use ability is acquired by SOTA LLMs remains unclear. In this paper, we aim to bridge this gap and fathom the underlying mechanism.

**Instruction Tuning** Instruction tuning enhances LLMs in understanding human instructions and generating proper responses (Wei et al., 2021; Bach et al., 2022). Since manual annotation is time-consuming, self-instruct (Wang et al., 2022) proposes to generate high-quality data from SOTA LLMs, which facilitates a recent trend of data curation for multi-turn dialogue (Taori et al., 2023; Chiang et al., 2023; Xu et al., 2023a; Ding et al., 2023). Compared with the dialogue, tool learning is more challenging given the vast diversity of APIs and the complexity of multi-tool instructions. As a result, even GPT-4 often fails to find a valid solution path. However, the existing tool-learning dataset cannot effectively address real human needs as mentioned in § 1. Instead, ToolBench is designed for practical scenarios and improves the previous pipeline for tool-learning data construction.

**Prompting LLMs for Decision Making** Prompting facilitates LLMs to decompose high-level tasks into sub-tasks and generate grounded plans (Ahn et al., 2022; Huang et al., 2022a;b; Ye et al., 2023). ReACT (Yao et al., 2022) integrates reasoning with acting by allowing LLMs to give a proper reason for an action and incorporating environmental feedback for reasoning. However, these studies do not incorporate a mechanism for decision retraction, which becomes problematic as an initial error can lead to a cascade of subsequent errors. Recently, Reflexion (Shinn et al., 2023) mitigates this issue by asking LLMs to reflect on previous failures. Our DFSDT extends Reflexion to a more general method by allowing LLMs to assess different reasoning paths and select the most promising one. In essence, DFSDT shares a similar idea to one concurrent work: tree-of-thought (ToT) reasoning (Yao et al., 2023). However, DFSDT targets general decision-making problems where the decision space is *infinite*, compared to ToT's relatively simple tasks that can be addressed by brute-force search.

## 5 CONCLUSION

To elicit the tool-use capabilities within LLMs, we present ToolBench, covering 16k+ real-world APIs and various practical use-case scenarios including both single-tool and multi-tool tasks. Moreover, we propose DFSDT to reinforce the planning and reasoning ability of LLMs, enabling them to navigate through reasoning paths strategically. For efficient evaluation of tool learning, we devise an automatic evaluator ToolEval. By fine-tuning LLaMA on ToolBench, the obtained model ToolLLaMA matches the performance of ChatGPT and exhibits remarkable generalization ability to unseen APIs. Besides, we develop a neural API retriever to recommend relevant APIs for each instruction. The retriever can be integrated with ToolLLaMA as a more automated tool-use pipeline. In the experiments, we demonstrate the generalization ability of our pipeline to out-of-distribution domains. In general, this work paves the way for future research in the intersection of instruction tuning and tool use for LLMs.

## ACKNOWLEDGEMENTS

The contributions are listed as follows: (1) API collection: Shihao Liang, Sihan Zhao, Kunlun Zhu, Yujia Qin; (2) instruction generation: Lan Yan, Kunlun Zhu, Shihao Liang, Yujia Qin; (3) solution path annotation: Yining Ye, Shihao Liang, Runchu Tian, Yujia Qin, Xin Cong; (4) model implementation: Shihao Liang, Yujia Qin, Kunlun Zhu, Lauren Hong, Yifan Wu; (5) system demonstration: Xiangru Tang, Bill Qian. Yujia Qin led the project, designed the methodology and experiments, and wrote the paper. Yankai Lin, Mark Gerstein, Dahai Li, Zhiyuan Liu, Maosong Sun, and Jie Zhou advised the project. Yankai Lin, Xin Cong, and Ruobing Xie proofread the whole paper. All authors participated in the discussion. Yujia Qin is sponsored by the Baidu Scholarship.

The authors would like to thank Yifan Wu, Si Sun, Zheni Zeng, Chen Zhang, Yu Gu, Chenfei Yuan, Junxi Yan, Shizuo Tian, Mingxi Yan, Jason Phang, Chen Qian, and Weize Chen for their valuable feedback, discussion, and participation in this project.

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

APPENDIX

## A    IMPLEMENTATION DETAILS

### A.1    DETAILS FOR FILTERING RAPIDAPI

We perform a rigorous filtering process to ensure that the ultimate tool set of ToolBench is reliable and functional. The filtering process is as follows: (1) *initial testing*: we begin by testing the basic functionality of each API to ascertain whether they are operational. We discard any APIs that do not meet this basic criterion; (2) *example response evaluation*: we make API calls to obtain an example response. Then we evaluate their effectiveness by response time and quality. APIs that consistently exhibit a long response time are omitted. Also, we filter out the APIs with low-quality responses, such as HTML source codes or other error messages.

### A.2    API RESPONSE COMPRESSION

When examining the response returned by each API, we discover that some responses may contain redundant information and are too long to be fed into LLMs. This may lead to problems due to the limited context length of LLMs. Therefore, we perform a response compression to reduce the length of API responses while maintaining their critical information.

Since each API has a fixed response format, we use ChatGPT to analyze one response example and remove unimportant keys within the response to reduce its length. The prompt of ChatGPT contains the following information for each API: (1) tool documentation, which includes tool name, tool description, API name, API description, parameters, and an example API response. This gives ChatGPT a hint of the API's functionality; (2) 3 in-context learning examples, each containing an original API response and a compressed response schema written by experts. In this way, we obtain the response compression strategies for all APIs. During inference, when the API response length exceeds 1024 tokens, we compress the response by removing unimportant information. If the compressed response is still longer than 1024, we only retain the first 1024 tokens. Through human evaluation, we find that this compression retains important information contained in the API response and successfully removes the noises.

### A.3    DETAILS FOR TRAINING TOOLLLAMA

We train the model in a multi-round conversation mode. For the training data format, we keep the input and output the same as those of ChatGPT. Since it is unclear how ChatGPT organizes the function call field, we just concatenate this information into the input as part of the prompt for ToolLLaMA. For the training hyper parameters, we use a learning rate of $5 \times 10^{-5}$, a warmup ratio of $4 \times 10^{-2}$, a total batch size of $64$, a maximum sequence length of $8192$, and use a position interpolation ratio of 2. We train the model for two epochs and select the model checkpoint with the best performance on the development set and then evaluate it on the test set.

### A.4    DETAILS FOR DFSDT

In practice, it is essential to balance effectiveness with costs (the number of OpenAI API calls). Classical DFS algorithms generate multiple child nodes at each step, then sort all the child nodes, and select the highest-scoring node for expansion. After greedily expanding to the terminal node, DFS backtracks to explore nearby nodes, expanding the search space. Throughout the algorithm, the most resource-intensive part is the sorting process of child nodes. If we use an LLM to evaluate two nodes at a time, it requires approximately $O(n \log n)$ complexity of OpenAI API calls, where $n$ is the number of child nodes.

In fact, we find empirically that in most cases, the nodes ranked highest are often the node generated at first. Therefore, we skip the sorting process of child nodes and choose a pre-order traversal (a variant for DFS) for the tree search. This design has the following advantages:

- If the model does not retract an action (e.g., for the case of simple instructions), then DFSDT degrades to ReACT, which makes it as efficient as ReACT.

- After the algorithm finishes, the nodes explored by this method are almost the same as those found by a classical DFS search. Hence, it can also handle complex instructions that only DFS can solve.

Overall, this design achieves a similar performance as DFS while significantly reducing costs.

It should also be noted that ReACT can be viewed as a degraded version of DFSDT. Therefore, although ToolLLaMA is trained on data created by DFSDT, the model can be used either through ReACT or DFSDT during inference.

## A.5 DETAILS FOR TOOLEVAL

We adopt two metrics for automatic tool-use capability evaluation: pass rate and win rate.

**Details for Pass Rate**   To assess whether a solution path completes the tasks outlined in the original instruction and successfully passes it, we need to first consider the solvability of the instruction. In principle, an instruction can be classified as either (1) solvable: for example, at least one of the provided tools is potentially helpful in solving the original instruction; or (2) unsolvable: for example, all APIs are irrelevant to the instruction or the instruction provides invalid information such as invalid email address.

To determine whether a solution path is deemed passed or not, we need to consider whether the instruction is solvable or unsolvable. In our evaluation, three types of labels can be given to each solution path, i.e., `Pass`, `Fail`, and `Unsure`. Specifically, we define different rules as follows:

If the instruction is solvable:

1. If the model gives finish type "Finish by Giving Up",
   (a) After trying all the APIs extensively during and receiving no helpful information from APIs, the solution path is deemed a `Pass`.
   (b) If the model only calls a few API or receiving valid information from the APIs, the solution path is deemed a `Fail`.
2. If the model gives finish type "Finish with Final Answer",
   (a) If the APIs provide no valid information, and the model has tried all the APIs to retrieve useful information, but the final answer still does not resolve the original instruction or conveys a refusal (such as "I'm sorry, but I can't provide you with this, because the tools are unavailable"), the solution path is deemed a `Pass`.
   (b) If the tools provide valid information, and the final answer does not completely resolve the instruction or is a refusal, the solution path is deemed a `Fail`.
   (c) If the final answer completely resolves the original instruction, the solution path is deemed a `Pass`.
   (d) If it is unable to determine if the instruction is resolved based on the content of the final answer, the solution path is deemed an `Unsure`.

If the instruction is unsolvable:

1. If the model gives finish type "Finish with Final Answer",
   (a) If the final answer resolves an instruction that was initially considered unresolvable, the solution path is deemed a `Pass`.
   (b) If the final answer is a refusal, the solution path is deemed a `Pass`.
   (c) If the final answer is hallucinated by the model itself and provides a false positive response (such as "I've completed the task, the final answer is *"), the solution path is deemed a `Fail`.
2. If the model gives finish type "Finish by Giving Up",
   (a) Under this case, the solution path is deemed a `Pass`.

For every solution path, we instruct the ChatGPT evaluator to generate multiple ($\geq 4$) predictions and perform a majority vote to derive the final pass rate.

**Details for Win Rate** Since pass rate only measures whether an instruction is completed or not, instead of how well it is completed, we adopt another metric: win rate. It is measured by comparing two solution paths for a given instruction. We assume that a passed candidate is better than a failed candidate and only compare those solution paths that are both "`Pass`", or both "`Failed`" annotated by the ChatGPT evaluator. Note that compared with another solution path, one solution path will be annotated with one of the following: `win`, `lose`, or `tie`. We build rules for the evaluator's behavior to decide which solution path is better, and the criteria are listed as follows:

1. Information richness: whether the final answer contains all the necessary information to answer the original instruction. A significantly richer answer is better, while a similar level of richness that is sufficient to answer the question ties.

2. Factuality: whether it accurately describes what has been done, and what failed in the end. A more accurate description in the final answer is better.

3. Reasoning: whether a detailed and accurate reason for failure is provided if the query remains unresolved. A more detailed reason is better.

4. Milestone: calculating the number of milestones reached during execution.

5. Exploration: whether more potentially useful APIs were attempted during the execution process. The use of a greater number of APIs is better.

6. Cost: Having fewer repeated (redundant) API calls is better if the number of APIs used is the same.

For every solution path, we also generate multiple ($\geq 4$) predictions and then perform a majority vote to derive the final win rate. In Table 4, for ease of reading, we split the ratio of `tie` into two pieces and add them to `win` and `lose`, respectively. In Table 6, we report the original numbers as a reference.

**Comparing Human Evaluation and ToolEval** To validate the reliability of ChatGPT evaluator in both pass rate and win rate, we sample among four different methods (ChatGPT+ReACT, ChatGPT+DFSDT, ToolLLaMA+DFSDT and GPT4+DFSDT) to obtain solution pairs for 300 test instructions for **each** method. Then we engage humans to annotate the pass rate for ChatGPT+DFSDT, ToolLLaMA+DFSDT and GPT4+DFSDT, and the win rate among ChatGPT+ReACT and ChatGPT+DFSDT. Our ChatGPT evaluator demonstrates a high agreement of **87.1%** in pass rate and **80.3%** in win rate with human annotators. This result shows that our evaluator generates highly similar evaluation results to humans and can be viewed as a credible evaluator who simulates human evaluation on pass rate and win rate.

It should also be noted that the evaluation for tool learning is far more intricate than traditional tasks such as dialogue. The reason is that there may exist infinite "correct" solution paths for each instruction. In our initial investigations, we surprisingly found that even human experts often disagree with each other in deciding which solution path is better, leading to a relatively low agreement. For instance, one may prefer a solution path that uses only a few APIs to derive the final answer quickly; while another may prefer a solution path that extensively tries all the APIs to cross-validate specific information. In this regard, we believe there is still a long way to go for a fair evaluation of the tool-use domain, and we believe this work has paved the way for it. We expect more future works to explore this interesting research problem.

## A.6  DETAILS FOR EXPERIMENTS ON APIBENCH

When generalizing ToolLLaMA to APIBench, no training updates were made to ToolLLaMA, but instead of treating each API in the prompt as a function call. We define one function that represents selecting an API, providing the code for invoking it, and describing the generated output in natural language. We do not consider the zero-shot setting of APIBench where the prompts do not contain any API descriptions because the APIs from the three tested domains were never encountered during training.

| Model | Method | I1-Inst. | | I1-Tool | | I1-Cat. | | I2-Inst. | | I2-Cat. | | I3-Inst. | | Average | |
|---|---|---|---|---|---|---|---|---|---|---|---|---|---|---|---|
| | | Win | Tie | Win | Tie | Win | Tie | Win | Tie | Win | Tie | Win | Tie | Win | Tie |
| ChatGPT | DFSDT | 52.5 | 16.0 | 55.0 | 14.0 | 47.5 | 19.5 | 67.0 | 10.0 | 58.5 | 12.5 | 61.0 | 16.0 | 56.9 | 14.7 |
| Claude-2 | ReACT | 27.0 | 8.0 | 24.0 | 7.5 | 29.5 | 8.5 | 32.0 | 6.0 | 28.5 | 6.0 | 43.0 | 9.5 | 30.7 | 7.5 |
| | DFSDT | 34.0 | 8.0 | 41.0 | 6.5 | 39.5 | 7.5 | 32.5 | 9.5 | 33.5 | 0.0 | 65.0 | 0.0 | 40.8 | 5.3 |
| Text-Davinci-003 | ReACT | 23.5 | 10.0 | 28.5 | 13.5 | 27.0 | 8.0 | 26.5 | 6.5 | 25.5 | 8.5 | 41.0 | 8.0 | 28.7 | 9.1 |
| | DFSDT | 35.0 | 10.5 | 37.5 | 12.5 | 40.0 | 13.5 | 36.5 | 8.0 | 40.0 | 6.5 | 60.0 | 6.0 | 41.5 | 9.5 |
| GPT4 | ReACT | 52.5 | 15.0 | 53.5 | 10.5 | 56.0 | 15.0 | 59.5 | 12.5 | 52.5 | 15.5 | 76.0 | 4.0 | 58.3 | 12.1 |
| | DFSDT | 60.5 | 14.0 | 62.5 | 10.5 | 58.0 | 17.0 | 67.0 | 12.5 | 57.0 | 12.5 | 80.0 | 8.0 | 64.2 | 12.4 |
| Vicuna | (ReACT & DFSDT) | 0.0 | 0.0 | 0.0 | 0.0 | 0.0 | 0.0 | 0.0 | 0.0 | 0.0 | 0.0 | 0.0 | 0.0 | 0.0 | 0.0 |
| Alpaca | (ReACT & DFSDT) | 0.0 | 0.0 | 0.0 | 0.0 | 0.0 | 0.0 | 0.0 | 0.0 | 0.0 | 0.0 | 0.0 | 0.0 | 0.0 | 0.0 |
| ToolLLaMA | ReACT | 40.0 | 10.0 | 36.5 | 11.0 | 42.0 | 11.0 | 45.5 | 10.5 | 37.5 | 8.5 | 51.0 | 8.0 | 42.1 | 9.8 |
| | DFSDT | 48.5 | 13.0 | 50.5 | 9.5 | 49.5 | 10.0 | 62.5 | 12.0 | 52.0 | 12.0 | 68.0 | 2.0 | 55.2 | 9.8 |
| | Retriever | 58.0 | 8.5 | 54.5 | 9.0 | 51.0 | 8.0 | 64.5 | 8.0 | 56.0 | 9.5 | 71.0 | 4.0 | 59.2 | 7.8 |

Table 6: Win rate results before merging the tie label. Win rate is calculated by comparing each model with ChatGPT-ReACT. A win rate higher than $50\%$ means the model performs better than ChatGPT-ReACT. Apart from ToolLLaMA-DFSDT-Retriever, all methods use the oracle API retriever (i.e., ground truth API).

## A.7 PROMPTS FOR INSTRUCTION GENERATION

Below we list the detailed prompt for instruction generation, which consists of four parts: task description, in-context learning examples, sampled API list, and other requirements.

---

*Task Description of Single-tool Instructions*:
You will be provided with a tool, its description, all of the tool's available API functions, the descriptions of these API functions, and the parameters required for each API function. Your task involves creating 10 varied, innovative, and detailed user queries that employ multiple API functions of a tool. For instance, if the tool 'climate news' has three API calls - 'get_all_climate_change_news', 'look_up_climate_today', and 'historical_climate', your query should articulate something akin to: first, determine today's weather, then verify how often it rains in Ohio in September, and finally, find news about climate change to help me understand whether the climate will change anytime soon. This query exemplifies how to utilize all API calls of 'climate news'. A query that only uses one API call will not be accepted. Additionally, you must incorporate the input parameters required for each API call. To achieve this, generate random information for required parameters such as IP address, location, coordinates, etc. For instance, don't merely say 'an address', provide the exact road and district names. Don't just mention 'a product', specify wearables, milk, a blue blanket, a pan, etc. Don't refer to 'my company', invent a company name instead. The first seven of the ten queries should be very specific. Each single query should combine all API call usages in different ways and include the necessary parameters. Note that you shouldn't ask 'which API to use', rather, simply state your needs that can be addressed by these APIs. You should also avoid asking for the input parameters required by the API call, but instead directly provide the parameter in your query. The final three queries should be complex and lengthy, describing a complicated scenario where all the API calls can be utilized to provide assistance within a single query. You should first think about possible related API combinations, then give your query. Related_apis are apis that can be used for a give query; those related apis have to strictly come from the provided api names. For each query, there should be multiple related_apis; for different queries, overlap of related apis should be as little as possible. Deliver your response in this format: [Query1: ......, 'related_apis':[api1, api2, api3...],Query2: ......, 'related_apis':[api4, api5, api6...],Query3: ......, 'related_apis':[api1, api7, api9...], ...]

---

*Task Description of Multi-tool Instructions*:
You will be provided with several tools, tool descriptions, all of each tool's available API functions, the descriptions of these API functions, and the parameters required for each API function. Your task involves creating 10 varied, innovative, and detailed user queries that employ API functions of multiple tools. For instance, given three tools 'nba_news', 'cat-facts', and 'hotels': 'nba_news' has API functions 'Get individual NBA source news' and 'Get all NBA news', 'cat-facts' has API functions 'Get all facts about cats' and 'Get a random fact about cats', 'hotels' has API functions 'properties/get-details (Deprecated)', 'properties/list (Deprecated)' and 'locations/v3/search'. Your query should articulate something akin to: 'I want to name my newborn cat after Kobe and host a

party to celebrate its birth. Get me some cat facts and NBA news to gather inspirations for the cat name. Also, find a proper hotel around my house in Houston Downtown for the party.' This query exemplifies how to utilize API calls of all the given tools. A query that uses API calls of only one tool will not be accepted. Additionally, you must incorporate the input parameters required for each API call. To achieve this, generate random information for required parameters such as IP address, location, coordinates, etc. For instance, don't merely say 'an address', provide the exact road and district names. Don't just mention 'a product', specify wearables, milk, a blue blanket, a pan, etc. Don't refer to 'my company', invent a company name instead. The first seven of the ten queries should be very specific. Each single query should combine API calls of different tools in various ways and include the necessary parameters. Note that you shouldn't ask 'which API to use', rather, simply state your needs that can be addressed by these APIs. You should also avoid asking for the input parameters required by the API call, but instead directly provide the parameters in your query. The final three queries should be complex and lengthy, describing a complicated scenario where all the provided API calls can be utilized to provide assistance within a single query. You should first think about possible related API combinations, then give your query. Related APIs are APIs that can be used for a given query; those related APIs have to strictly come from the provided API names. For each query, there should be multiple related APIs; for different queries, overlap of related APIs should be as little as possible. Deliver your response in this format: [Query1: ......, 'related_apis':[[tool name, api name], [tool name, api name], [tool name, api name]...],Query2: ......, 'related_apis':[[tool name, api name], [tool name, api name], [tool name, api name]...],Query3: ......, 'related_apis':[[tool name, api name], [tool name, api name], [tool name, api name]...], ...]

---

*In-context Seed Examples.* In the following, we show one single-tool instruction seed example and one multi-tool instruction seed example.

For example, with tool ASCII Art, the given api_names are 'figlet', 'list figlet styles', 'cowsay', 'list_cowsay_styles', 'matheq'.
Some sample queries and related_apis would be:
"Query": "Need to create an ASCII art representation of a mathematical equation. The equation is 'y = mx + c', where m and c are constants. Help me generate the ASCII art for this equation. Also please generate an ASCII art representation of the text 'Newton's Second Law of Motion'.", "related_apis": ['figlet', 'list figlet styles', 'matheq']
"Query": "Working on a research paper on cows and need to include ASCII art representations of various cows. Can you first retrieve available ASCII art styles for cows? Then, can you generate ASCII art for cows like the Jersey, Holstein, and Guernsey? Finally, I want the cow to say 'Moo!' in the ASCII art.", "related_apis": ['figlet', 'list figlet styles', 'cowsay', 'list_cowsay_styles']
"Query": "I'm writing a blog post on ASCII art and need to include some examples. Can you generate ASCII art for the following strings: 'ASCII', 'art', and 'gallery'? You can first retrieve available figlet styles and then generate ASCII art for the strings using the styles.", "related_apis": ['figlet', 'list figlet styles']
"Query": "Greetings! I'm putting together a quirky slideshow about our furry friends and need your help to sprinkle some ASCII art goodness. Could you kindly fetch me the catalog of ASCII art styles available for animals? Also, I'm particularly keen on featuring ASCII art for creatures like pandas, cows, elephants, and penguins. And if they could say something cute like 'Hello!' or 'Hugs!' in the ASCII art, that would be purr-fect!", "related_apis": ['figlet', 'list figlet styles', 'cowsay', 'list_cowsay_styles']

For example, with tool ['Entrepreneur Mindset Collection', 'Random Words', 'thedigitalnewsfeederapi', 'Chemical Elements'], the given api_names are (tool 'Entrepreneur Mindset Collection')'Random Quote in JSON format', (tool 'Random Words')'Get multiple random words', (tool 'Random Words')'Get a random word', (tool 'thedigitalnewsfeederapi')'getting specific cricket articles', (tool 'thedigitalnewsfeederapi')'Getting Cricket Articles', (tool 'thedigitalnewsfeederapi')'getting specific news articles', (tool 'thedigitalnewsfeederapi')'Getting News Articles', (tool 'thedigitalnewsfeederapi')'getting all news articles', (tool 'Chemical Elements')'Get All Chemical Elements'.
Some sample queries and related_apis would be:
"Query": "For my best friend's surprise birthday party, I require inspiration for party games and decorations. Kindly suggest some random words that can serve as themes for the party. Furthermore, I'm interested in gathering news articles about the latest party trends to ensure a modern celebration.

Also, I would appreciate details about the local hotels in my area for accommodation options. Your assistance is greatly appreciated.", "related_apis": [['Random Words', 'Get multiple random words'], ['thedigitalnewsfeederapi', 'Getting News Articles'], ['thedigitalnewsfeederapi', 'Getting all news articles']]

"Query": "In the midst of organizing a team-building event for my esteemed company, I eagerly seek your valued input for invigorating activities. Might I kindly request a collection of random quotes that encapsulate the essence of teamwork and motivation? Additionally, I am keen on exploring news articles that showcase triumphant team-building events, as they serve as a wellspring of inspiration.", "related_apis": [['Entrepreneur Mindset Collection', 'Random Quote in JSON format'], ['thedigitalnewsfeederapi', 'Getting News Articles']] "Query": "I need specific cricket articles that discuss the health benefits of sports for my research paper on exercise. I also want to know which chemical elements are associated with exercising, like increased iron (Fe) and its impact on bone marrow.", "related_apis": [['thedigitalnewsfeederapi', 'getting specific cricket articles'], ['Chemical Elements', 'Get All Chemical Elements']]

"Query": "I'm starting a new business venture and I need to make a speech announcing the new dawn. Provide me some quotes and words for me to start with. I would like to gather news articles about successful entrepreneurs for inspiration.", "related_apis": [['Entrepreneur Mindset Collection', 'Random Quote in JSON format'], ['Random Words', 'Get multiple random words'], ['thedigitalnewsfeederapi', 'getting specific news articles']]

These are only examples to show you how to write the query. Do not use APIs listed in the above examples, but rather, use the ones listed below in the INPUT.

---

*Sampled API List* (An example)

```
{
    "tool_description": "EntreAPI Faker is used to dynamically
        create mock, demo, test and sample data for your
        application",
    "name": "EntreAPI Faker",
    "api_list": [
        {
            "name": "Longitute",
            "url": "https://entreapi-faker.p.rapidapi.com/address/
                longitude",
            "description": "Generate a random longitude.",
            "method": "GET",
            "required_parameters": [],
            "optional_parameters": [
                {
                    "name": "max",
                    "type": "NUMBER",
                    "description": "Maximum value for latitude.",
                    "default": ""
                },
                {
                    "name": "min",
                    "type": "NUMBER",
                    "description": "Minimum value for latitude.",
                    "default": ""
                },
                {

                    "name": "precision",
                    "type": "NUMBER",
                    "description": "Precision for latitude.",
                    "default": ""
                }
            ],
            "tool_name": "EntreAPI Faker",
            "category_name": "Data"
```

```
        },
        {
            "name": "Boolean",
            "url": "https://entreapi-faker.p.rapidapi.com/datatype
                /boolean",
            "description": "Randomly generate a boolean value.",
            "method": "GET",
            "required_parameters": [],
            "optional_parameters": [],
            "tool_name": "EntreAPI Faker",
            "category_name": "Data"
        },
        {

            "name": "Past",
            "url": "https://entreapi-faker.p.rapidapi.com/date/
                past",
            "description": "Randomly generate a date value in the
                past.",
            "method": "GET",
            "required_parameters": [],
            "optional_parameters": [
                {
                    "name": "refDate",
                    "type": "STRING",
                    "description": "Starting reference date",
                    "default": ""
                },
                {
                    "name": "years",
                    "type": "NUMBER",
                    "description": "Number of years for the range
                        of dates.",
                    "default": ""
                }
            ],
            "tool_name": "EntreAPI Faker",
            "category_name": "Data"
        },
        {

            "name": "Image Url",
            "url": "https://entreapi-faker.p.rapidapi.com/image/
                imageUrl",
            "description": "Randomly generate an image URL.",
            "method": "GET",
            "required_parameters": [],
            "optional_parameters": [
                {
                    "name": "width",
                    "type": "NUMBER",
                    "description": "Width of the image. Default is
                        640.",
                    "default": ""
                },
                {
                    "name": "height",
                    "type": "NUMBER",
                    "description": "Height of the image. Default
                        is 480.",
                    "default": ""
```

```
            },
            {
                "name": "useRandomize",
                "type": "BOOLEAN",
                "description": "Add a random number parameter
                    to the returned URL.",
                "default": ""
            },
            {
                "name": "category",
                "type": "STRING",
                "description": "The category for the image.
                    Can be one: abstract, animal, avatar,
                    business, cats, city, fashion, food,
                    nature, nightlife, people, sports,
                    technics, transport",
                "default": ""
            }
        ],
        "tool_name": "EntreAPI Faker",
        "category_name": "Data"
    },
    {
        "name": "Sentence",
        "url": "https://entreapi-faker.p.rapidapi.com/lorem/
            sentence",
        "description": "Randomly generate a sentence of Lorem
            Ipsum.",
        "method": "GET",
        "required_parameters": [],
        "optional_parameters": [
            {
                "name": "wordCount",
                "type": "NUMBER",
                "description": "Number of words in the
                    sentence.",
                "default": ""
            }
        ],
        "tool_name": "EntreAPI Faker",
        "category_name": "Data"
    },
    {
        "name": "Gender",
        "url": "https://entreapi-faker.p.rapidapi.com/name/
            gender",
        "description": "Randomly select a gender.",
        "method": "GET",
        "required_parameters": [],
        "optional_parameters": [
            {
                "name": "useBinary",
                "type": "BOOLEAN",
                "description": "Use binary genders only.",
                "default": ""
            }
        ],
        "tool_name": "EntreAPI Faker",
        "category_name": "Data"
```

```
        },
        {
            "name": "Prefix",
            "url": "https://entreapi-faker.p.rapidapi.com/name/
                prefix",
            "description": "Randomly generate a prefix (e.g., Mr.,
                 Mrs., etc.)",
            "method": "GET",
            "required_parameters": [],
            "optional_parameters": [
                {
                    "name": "gender",
                    "type": "STRING",
                    "description": "Optional gender.",
                    "default": ""
                }
            ],
            "tool_name": "EntreAPI Faker",
            "category_name": "Data"
        },
        {
            "name": "Array Element",
            "url": "https://entreapi-faker.p.rapidapi.com/random/
                arrayElement",
            "description": "Randomly select an array element.",
            "method": "GET",
            "required_parameters": [],
            "optional_parameters": [
                {
                    "name": "array",
                    "type": "ARRAY",
                    "description": "The list of elements to choose
                         from. Default is [\"a\", \"b\", \"c\"].",
                    "default": ""
                }
            ],
            "tool_name": "EntreAPI Faker",
            "category_name": "Data"
        },
        {
            "name": "Number Value",
            "url": "https://entreapi-faker.p.rapidapi.com/random/
                number",
            "description": "Randomly generate a number value.",
            "method": "GET",
            "required_parameters": [],
            "optional_parameters": [
                {
                    "name": "min",
                    "type": "NUMBER",
                    "description": "Minimum value.",
                    "default": ""
                },
                {
                    "name": "max",
                    "type": "NUMBER",
                    "description": "Maximum value.",
                    "default": ""
                },
```

```
                    {
                        "name": "precision",
                        "type": "NUMBER",
                        "description": "Precision of the number.",
                        "default": ""
                    }
                ],
                "tool_name": "EntreAPI Faker",
                "category_name": "Data"
            },
            {
                "name": "URL",
                "url": "https://entreapi-faker.p.rapidapi.com/internet
                    /url",
                "description": "Randomly generate a URL.",
                "method": "GET",
                "required_parameters": [],
                "optional_parameters": [],
                "tool_name": "EntreAPI Faker",
                "category_name": "Data"
            }
        ]
    }
```

---

*Other Requirements:*
Please produce ten queries in line with the given requirements and inputs. These ten queries should display a diverse range of sentence structures: some queries should be in the form of imperative sentences, others declarative, and yet others interrogative. Equally, they should encompass a variety of tones, with some being polite, others straightforward. Ensure they vary in length and contain a wide range of subjects: myself, my friends, family, and company. Aim to include a number of engaging queries as long as they relate to API calls. Keep in mind that for each query, invoking just one API won't suffice; each query should call upon two to five APIs. However, try to avoid explicitly specifying which API to employ in the query. Each query should consist of a minimum of thirty words.

---

## A.8    PROMPTS FOR SOLUTION PATH ANNOTATION

We use the following prompt when searching for the solution path. When expanding the child nodes, we use diversity_user_prompt, showing the information of previous child nodes.

```
----------------------------------------------------------------
system_prompt:
You are Tool-GPT, capable of utilizing numerous tools and
   functions to complete the given task.
1.First, I will provide you with the task description, and your
   task will commence.
2.At each step, you need to analyze the current status and
   determine the next course of action by executing a function
   call.
3.Following the call, you will receive the result, transitioning
   you to a new state. Subsequently, you will analyze your
   current status, make decisions about the next steps, and
   repeat this process.
4.After several iterations of thought and function calls, you will
    ultimately complete the task and provide your final answer.
Remember:
1.The state changes are irreversible, and you cannot return to a
   previous state.
```

2. Keep your thoughts concise, limiting them to a maximum of five
   sentences.
3. You can make multiple attempts. If you plan to try different
   conditions continuously, perform one condition per try.
4. If you believe you have gathered enough information, call the
   function "Finish: give_answer" to provide your answer for the
   task.
5. If you feel unable to handle the task from this step, call the
   function "Finish: give_up_and_restart".
Let's Begin!
Task description: {task_description}
----------------------------------------------------------
diversity_user_prompt:
This is not the first time you try this task, all previous trails
   failed.
Before you generate your thought for this state, I will first show
    you your previous actions for this state, and then you must
   generate actions that is different from all of them. Here are
   some previous actions candidates:
{previous_candidate}
Remember you are now in the intermediate state of a trail, you
   will first analyze the now state and previous action
   candidates, then make actions that is different from all the
   previous.
----------------------------------------------------------
Finish_function_description:
{
    "name": "Finish",
    "description": "If you believe that you have obtained a result
        that can answer the task, please call this function to
       provide the final answer. Alternatively, if you recognize
       that you are unable to proceed with the task in the
       current state, call this function to restart. Remember:
       you must ALWAYS call this function at the end of your
       attempt, and the only part that will be shown to the user
       is the final answer, so it should contain sufficient
       information.",
    "parameters": {
        "type": "object",
        "properties": {
            "return_type": {
                "type": "string",
                "enum": ["give_answer","give_up_and_restart"],
            },
            "final_answer": {
                "type": "string",
                "description": "The final answer you want to give
                    the user. You should have this field if \"
                    return_type\"==\"give_answer\"",
            }
        },
        "required": ["return_type"],
    }
}

