# OpenReview forum: "ToolLLM: Facilitating Large Language Models to Master 16000+ Real-world APIs"
_ICLR.cc/2024/Conference — ICLR 2024 spotlight_

### Official Review · Reviewer_u7K9 · 2023-10-26

**Soundness:** 3 good
**Presentation:** 3 good
**Contribution:** 3 good
**Rating:** 6
**Confidence:** 5

**Summary:**

Tool utilization is an important capability of LLMs to extend their task scope. Although closed-source LLMs have achieved powerful performance by calling external tools, it is still important for open-source LLMs to enhance their capability in tool use. In this paper, the authors proposed ToolLLM to facilitate the large language model to master 16,000+ tools. Moreover, the contributions of this paper can be summarized in three parts: 1) ToolBench, a benchmark which generate instructions involving different tool utilization; 2) a depth-first search-based decision tree algorithm is introduced to enhance the capability of LLMs in tool utilization; 3) an evaluation platform called TaskEval. Experimental results demonstrate the effectiveness of the proposed method.

**Strengths:**

1. Although tool utilization has received much attention in LLM applications, building a standard benchmark is still challenging. To address this, this paper releases a high-quality instruction tuning dataset, called ToolBench, which covers many different tools.
2. Based on the constructed ToolBench, this paper also designs a strategy for solution path annotation, which uses a depth-first search-based decision tree to search for a possible valid path.
3. Experimental results validate that tuning LLMs with generated samples can effectively improve performance.
4. The writing of this paper is good and easy to follow.

**Weaknesses:**

1. This paper introduces TaskEval, which encompasses two metrics, *Pass Rate* and *Win Rate*. Specifically, *Pass Rate* detects whether LLM can successfully execute user instructions, and *Win Rate* is designed to judge which solution path is better for a given instruction. However, to some degree, I think these metrics still have some deficiencies and cannot efficiently the capability of LLMs in tool utilization. For example, *Pass Rate* can only reflect LLM whether can execute user instruction. Sometimes, powerful LLMs (e.g., ChatGPT) can always generate answers for any user instructions without any tool use, and besides, the hallucination of LLMs can also let it generate some executed but counterfactual answers. Besides, *Win Rate* can only reflect the capability of different LLMs, not their ability of tool utilization. Even two LLMs without tool utilization (e.g., ChatGPT v.s. LLaMA-7b), will also demonstrate differences in performance. Therefore, I think the proposed metrics can only reflect the capability of LLMs but not their tool-use ability. Of course, I also admit that it is challenging to build such a metric since there have been no metrics in this area before.
2. I appreciate that the paper releases a high-quality dataset in this area. However, it will be better to provide a detailed statistical of this dataset (e.g., distribution in domain, error analysis).

**Questions:**

1. In the design of solution path annotation, authors prefer to use DFS instead of BFS. However, DFS could also possibly backtrack and search multiple answers if it does not obtain a valid path. So, any experiments to prove the cost of DFS to find a valid path when compared with BFS?
2. This paper mainly uses RapidAPIs for training. So is the fine-tuned ToolLLaMa suitable or adapted for APIs in other sources/formats (e.g., Pytorch/Tensorflow/HuggingFace in Gorilla, ChatGPT plugins)?

---

> ### Author Response · Authors · 2023-11-16
> **Response - 1**
>
> Q1: This paper introduces TaskEval, which encompasses two metrics, Pass Rate and Win Rate. Specifically, Pass Rate detects whether LLM can successfully execute user instructions, and Win Rate is designed to judge which solution path is better for a given instruction. However, to some degree, I think these metrics still have some deficiencies and cannot efficiently the capability of LLMs in tool utilization. For example, Pass Rate can only reflect LLM whether can execute user instruction. Sometimes, powerful LLMs (e.g., ChatGPT) can always generate answers for any user instructions without any tool use, and besides, the hallucination of LLMs can also let it generate some executed but counterfactual answers. Besides, Win Rate can only reflect the capability of different LLMs, not their ability of tool utilization. Even two LLMs without tool utilization (e.g., ChatGPT v.s. LLaMA-7b), will also demonstrate differences in performance. Therefore, I think the proposed metrics can only reflect the capability of LLMs but not their tool-use ability. Of course, I also admit that it is challenging to build such a metric since there have been no metrics in this area before.
>
> A1: Thank you for your insightful comments regarding our ToolEval metrics, including Pass Rate and Win Rate. We appreciate your concerns about the potential limitations of these metrics in fully capturing the tool utilization capabilities of LLMs. Below are our detailed responses:
>
> + **Purpose and Scope of Pass Rate**: The Pass Rate metric is designed to assess whether an LLM can successfully execute user instructions. While it is true that this metric primarily focuses on the execution aspect, it is an essential baseline measure. It helps to filter out responses that are non-executable or irrelevant, which is a fundamental requirement for effective tool use. The concern about powerful LLMs like ChatGPT generating answers without tool use is valid. However, in our experimental setup, the emphasis is on the execution of tasks involving specific tools, thereby directly relating to tool utilization.
>
> + **Role of Win Rate in Evaluating Tool Utilization**: The Win Rate metric, designed to judge the quality of solution paths, indirectly assesses the tool utilization capabilities of LLMs. It compares different models based on the effectiveness of their solutions, which inherently involves the utilization of tools. While it’s true that Win Rate also reflects general LLM capabilities, in the context of tasks that require tool use, it becomes a measure of how well these tools are utilized in achieving the desired outcome. Win rate not only considers the final answer but also the solution path. The solution path also reflects LLM’s ability to use tools to solve problems.
>
> + **Pioneering Effort in a New Metric Space**: It is important to recognize that the development of metrics for evaluating LLMs in tool utilization is a relatively new and evolving area. Our ToolEval represents one of the first forays into this space. While not exhaustive, these metrics provide a foundational framework for assessing LLMs in this context. We are open to evolving these metrics and incorporating additional measures as the field progresses.
>
> + **Continuous Improvement and Future Work**: We are committed to refining our evaluation methods. Future iterations of our work will focus on developing more nuanced metrics that can better capture the specific nuances of tool utilization, possibly incorporating aspects like tool relevance, efficiency of use, and the complexity of tasks executed.
>
> In conclusion, while the ToolEval metrics have certain limitations, they represent a significant step forward in assessing LLMs' tool-use capabilities. As pioneers in this space, we acknowledge the need for continuous improvement and welcome further research to develop more comprehensive metrics. Our current metrics provide a valuable starting point for this evolving area of study.
>
> -----------
>
> Q2: I appreciate that the paper releases a high-quality dataset in this area. However, it will be better to provide a detailed statistical of this dataset (e.g., distribution in domain, error analysis).
>
> A2: Thank you for acknowledging the quality of the dataset we have released in our paper. We also appreciate your suggestion to provide a more detailed statistical analysis of the dataset, including domain distribution and error analysis. We understand the importance of offering comprehensive insights into our dataset, as detailed statistics can significantly enhance its utility and applicability for future research. Therefore, we are committed to including a thorough statistical analysis in our revised paper.

---

> ### Author Response · Authors · 2023-11-16
> **Response - 2**
>
> Q3: In the design of solution path annotation, authors prefer to use DFS instead of BFS. However, DFS could also possibly backtrack and search multiple answers if it does not obtain a valid path. So, any experiments to prove the cost of DFS to find a valid path when compared with BFS?
>
> A3: Thank you for your question concerning the use of Depth First Search (DFS) over Breadth First Search (BFS) in our solution path annotation approach. We appreciate the opportunity to address this query with reference to empirical evidence.
>
> In our research, the preference for DFS over BFS was primarily driven by the goal of maximizing efficiency in finding valid solution paths. While it is true that DFS may require backtracking when an invalid path is encountered, our implementation of DFS in the DFSDT method is specifically designed to minimize the cost and time associated with such backtracking. DFS is more easily to reach a plausible solution path than BFS, and is more efficient than the latter.
>
> To validate the efficiency of DFS in comparison with BFS, we refer to the results from a follow-up study to our work [1]. This independent study conducted a comprehensive comparison between two versions of Tree of Thoughts (ToT) - one using BFS and the other using DFS - and our DFSDT method, which is an improved version of DFS. Below is the results on three different benchmarks (Game of 24, WebShop, and our ToolBench):
>
> | model                   | Game of 24 | WebShop | IToolBench |
> |-------------------------|----------|----------|---------|
> | CoT@3 (self-consistency)  | 7.00       | 56.45       | 31.20      |
> | ToT (BFS)  | 11.00       | 50.20       | 38.00      |
> | ToT (DFS) | 14.00       | 55.60       | 45.58      |
> | DFSDT (Ours)               | 29.00       | 57.25       | 50.20      |
>
> The results from this comparison, as shown in the table provided, clearly demonstrate that **DFSDT outperforms both BFS and DFS implementations of ToT across multiple benchmarks**, including Game of 24, WebShop, and our own ToolBench. Specifically, DFSDT not only achieves higher accuracy but also does so with fewer API calls, indicating a more efficient path to finding valid solutions.
>
> Besides the performance, the authors of work [1] also show that our **DFSDT is far more efficient than ToT (BFS)** within limited API calls (Figure 2 in work [1]), demonstrating DFSDT’s excellent efficiency in finding a good solution path. For more detailed discussion on the difference between these works, please kindly refer to paper [1].
>
> [1] Anonymous: Rational Decision-Making Agent with Internalized Utility Judgment   https://openreview.net/forum?id=l1pNNQSzZv
>
>
> Q4: This paper mainly uses RapidAPIs for training. So is the fine-tuned ToolLLaMa suitable or adapted for APIs in other sources/formats (e.g., Pytorch/Tensorflow/HuggingFace in Gorilla, ChatGPT plugins)?
>
> A4: Thanks for your suggestions!  As shown in Table 5, to empirically validate ToolLLaMa's generalization capabilities, **we conducted extensive out-of-distribution (OOD) experiments using the APIBench dataset**, which includes diverse APIs from Pytorch, Tensorflow, and HuggingFace. These platforms, as mentioned, are different from the ones in our training set and present a robust testbed for evaluating our model's adaptability. The results of these experiments, as detailed in Table 5, demonstrate remarkable OOD generalization performance by ToolLLaMa. Specifically, ToolLLaMa, equipped with our API retriever, consistently outperforms the Gorilla model (fine-tuned on APIBench) in AST accuracy across different datasets like HuggingFace and TorchHub. This is a strong indicator of ToolLLaMa's ability to adapt to and perform well with APIs not present in its training data.
>
> Furthermore, when comparing ToolLLaMa with the same oracle retriever as Gorilla, our model consistently shows superior performance, particularly in zero-shot (ZS) scenarios. This further underscores ToolLLaMa's robustness and flexibility in handling diverse APIs, even those it was not explicitly trained on.
>
> In conclusion, the findings from our OOD experiments firmly establish **ToolLLaMa's generalization capabilities across various API sources and formats**. The model's strong performance in these tests reflects its adaptability and suitability for a wide range of APIs, extending well beyond the scope of its initial training set on RapidAPIs. This versatility is a key strength of ToolLLaMa, addressing your concerns about its applicability to different API platforms.

---

### Official Review · Reviewer_2o14 · 2023-10-30

**Soundness:** 4 excellent
**Presentation:** 4 excellent
**Contribution:** 3 good
**Rating:** 8
**Confidence:** 5

**Summary:**

- The paper introduces ToolLLM, a framework to facilitate tool use capabilities in open-source large language models (LLMs). It includes data construction, model training, and evaluation components.
- A new instruction tuning dataset called ToolBench is constructed using ChatGPT. It contains over 16,000 real-world APIs from RapidAPI spanning 49 categories. The dataset covers both single-tool and multi-tool instructions.
- An automatic evaluator ToolEval is developed to assess tool use capabilities. It incorporates pass rate to measure executability and win rate to compare solution quality.
- By fine-tuning LLaMA on ToolBench, ToolLLaMA is obtained. Experiments show it matches ChatGPT's performance and generalizes well to unseen APIs. An API retriever is also trained to automatically recommend relevant APIs. ToolLLaMA demonstrates strong generalization on the out-of-distribution APIBench dataset, despite not being trained on it. This validates its capabilities on new domains.

**Strengths:**

Leveraging existing techniques in the literature, authors build a framework for developing models capable of tool use. This framework encompasses dataset building, model training, and model evaluation. The scope is comprehensive, and the execution is generally solid.

The authors do a good job documenting and elaborating design decisions such as dataset filtering and issues with prompting with a limited context window.

The authors make their artifacts (ToolBench, ToolEval, and model artifacts) publicly available so others can build on their work.

**Weaknesses:**

Most of the technical ideas in this work are from past works, with the exception of DFSDT, which is a simple application of DFS to prompting. In this sense, the current work's technical contribution is low.

**Questions:**

NA

---

> ### Author Response · Authors · 2023-11-16
> **Response**
>
> Q: Most of the technical ideas in this work are from past works, with the exception of DFSDT, which is a simple application of DFS to prompting. In this sense, the current work's technical contribution is low.
>
> A: Thank you very much for your insightful comments. Below we wish to highlight several key aspects that distinguish our work:
>
> + **Limitation of Previous Works**: As mentioned in the introduction, existing works fail to fully stimulate the tool-use capabilities within LLMs and have inherent limitations: (1) **limited APIs**: they either fail to involve real-world APIs (e.g., RESTAPI) or consider only a small scope of APIs with poor diversity; (2) **constrained scenario**: existing works are confined to instructions that only involve one single tool. Besides, they often assume that users manually specify the ideal API set for a given instruction in advance, which is infeasible in practice; In addition, some works do not even execute APIs to obtain real responses, which serve as important information for subsequent model planning.
>
> + **Our Unique Setting and Rich Resources**: Different from exsiting works, our ToolBench, stands out due to its comprehensive and diverse collection of **over 16,000** REST APIs from RapidAPI. For each API, we crawl detailed API documents from RapidAPI, including the functionality descriptions, required parameters, code snippets for API calls, etc. **This level of diversity in dataset construction is unprecedented in previous studies**. We believe the resource we provide represents a significant advancement in the field of tool learning research. Its diversity and comprehensiveness offer a robust platform for future studies focused on enhancing tool-use capabilities in LLMs.
>
> + **Innovative Instruction Crafting Methodology**: Our methodology for crafting instructions is notably different from prior works. Instead of brainstorming instructions from scratch and then searching for relevant APIs, we adopt a reverse engineering approach. We sample different combinations of APIs and craft various instructions that involve them. This strategy ensures **comprehensive coverage for all collected APIs** and leads to the generation of instructions that are more aligned with real-world application scenarios.
>
> + **Leveraging RapidAPI Hierarchy for Multi-tool Instructions**: To address the challenge of creating realistic multi-tool instructions, we use the RapidAPI hierarchy to identify tool relationships. This aids in generating multi-tool instructions that are not only diverse but also functionally coherent. Our approach to generating intra-category and intra-collection multi-tool instructions is a novel strategy that effectively tackles the sparsity issue often encountered in random sampling of tool combinations.
>
> + **Function Call Feature Utilization**: We innovatively leverage the function call feature of ChatGPT by treating each API as a special function and feeding its documentation into ChatGPT's function field. To the best of our knowledge, we are **the first** to introduce function calling feature into the filed of tool learning. This approach fully exploits ChatGPT’s capability of tool use.
>
> + **Depth First Search-based Decision Tree (DFSDT)**: Existing works adopted either CoT or ReACT for model reasoning, which cannot fully elicit the capabilities stored in LLMs and thus fail to handle complex instructions in tool learning. To overcome the limitations of existing methods like CoT and ReACT, we propose a novel Depth First Search-based Decision Tree. This method allows ChatGPT to assess different reasoning paths, choose promising directions, or abandon ineffective ones. Our DFSDT approach significantly enhances the exploration of the action space, increasing the likelihood of finding valid solution paths. In experiments, DFSDT significantly improves the annotation efficiency and successfully completes those complex instructions that cannot be fulfilled using ReACT.
>
> + **Benchmarking Against Established Models**: To assess the tool-use capabilities of LLMs, we develop an automatic evaluator, ToolEval. We demonstrate that ToolEval achieves a high correlation with human evaluation and provides a robust, scalable, and reliable assessment for machine tool use. Our comparative analysis using ToolEval offers a new perspective on how LLaMA, fine-tuned on ToolBench, measures up against well-established models (Claude, Davinci, ChatGPT, GPT-4). This benchmarking is essential for understanding the relative progress in the field and provides a practical reference point for future research.
>
> + **A Capable Open-source Model Released**: We provide a capable model ToolLLaMA, which demonstrates a compelling capability to handle both single-tool and complex multi-tool instructions. ToolLLaMA outperforms Text-Davinci-003 and Claude-2, achieves comparable performance to the ChatGPT, and is only slightly inferior to GPT4. Also, ToolLLaMA exhibits robust generalization to previously unseen APIs.

---

> > ### Comment · Reviewer_2o14 · 2023-11-23
> > **acknowledged**
> >
> > I read the author comment and I am keeping my score.

---

### Official Review · Reviewer_k5uq · 2023-10-31

**Soundness:** 2 fair
**Presentation:** 3 good
**Contribution:** 2 fair
**Rating:** 6
**Confidence:** 4

**Summary:**

The paper collected a new instruction-tuning dataset (called ToolBench) for improving LLM's tool-use capability. To construct ToolBench, the author collected more than 16k REST APIs from RapidAPI and constructed synthesized instructions based on these APIs. To automatically obtain the desired behavior of the LLM, the author prompted prompt gpt-3.5-turbo-16k. By finetuning LLaMA on ToolBench, the author showed that ToolLLaMA achieves higher score according to a ChatGPT evaluator than Text-davinci-003 and Claude-2 in ToolBench.

**Strengths:**

The paper is valuable to those who are trying to build LLMs that can use tools. ToolBench is a well-engineered dataset and is shown to be more diverse than prior datasets in Table 1.

**Weaknesses:**

1. There are lots of papers that studied how to instruction-tune an LLM for tool-use, like GPT4Tools (in NeurIPS 2023), Gorilla, ToolAlpaca, etc. The techniques adopted in ToolLLM, such as constructing instruction-tuning samples by prompting ChatGPT, is very standard and has limited novelty. The paper is mainly about applying a popular synthetic instruction generation pipeline to distill knowledge from gpt-3.5 and has almost no research value.

2. The author proposed "Depth First Search-based Decision Tree" (DFSDT) that seems to outperform ReAct prompting. However, the technique is closely related to self-consistency + CoT and Tree-of-thoughts. It is also unclear how DFSDT will perform for other types of LLMs not in the category of LLaMA / GPT / Claude.

3. The evaluation metrics are based on ChatGPT, which is highly unreliable and may favor models tuned with ChatGPT-prompted datasets. The author mentioned in Appendix A.5 (Paragraph "Comparing Human Evaluation and ToolEval") that `Our ChatGPT evaluator demonstrates a high agreement of 87.1% in pass rate and 80.3% in win rate with human annotators. `. This shows that there is **close to 20% disagreement** between human evaluation and ChatGPT evaluation. This is a large discrepancy and should not be ignored.

**Questions:**

How does the models in Table 4 compare with each other if we adopt human evaluation?

---

> ### Author Response · Authors · 2023-11-15
> **Response - 1**
>
> Q1: There are lots of papers that studied how to instruction-tune an LLM for tool-use, like GPT4Tools (in NeurIPS 2023), Gorilla, ToolAlpaca, etc. The techniques adopted in ToolLLM, such as constructing instruction-tuning samples by prompting ChatGPT, is very standard and has limited novelty. The paper is mainly about applying a popular synthetic instruction generation pipeline to distill knowledge from gpt-3.5 and has almost no research value.
>
> A1-1: Thank you very much for your insightful comments. Our work is a concurrent endeavor, developed alongside your mentioned works, and focuses on a unique setting. This distinct approach, coupled with our technical innovations and the significant contribution of the ToolBench resource, underscores the novelty and importance of our study. Beyond technical advancements, our work provides broader insights into LLM technologies, affirming its value in the evolving landscape of LLM research. Below we wish to highlight several key aspects that distinguish our work:
>
> + **Concurrent Work**: Our ToolBench dataset was first open-sourced on **May 28, 2023** (on GitHub), which aligns closely with the release dates of GPT4Tools (**May 30, 2023**), Gorilla (**May 24, 2023**), and ToolAlpaca (**June 8, 2023**). This timeline indicates that our work was developed **concurrently** with these studies, rather than being a subsequent iteration of existing techniques. The concurrent development underscores that our approach was formulated independently and without the influence of these contemporaneous studies.
>
> Secondly, our work explores a **substantially different setting** from previous studies. While the basic concept of using ChatGPT for instruction-tuning samples may not be entirely new, the specific application and context in our study are novel. We focus on a unique aspect of leveraging large language models for tool-use capability enhancement. This setting addresses unexplored challenges and offers fresh insights into the capabilities and limitations of these models in practical scenarios. Besides, the value of the resources we provide is substantial. Specifically:
>
> + **Limitation of Previous Works**: As mentioned in the introduction, existing works (including GPT4Tools, Gorilla, and ToolAlpaca) fail to fully stimulate the tool-use capabilities within LLMs and have inherent limitations: (1) **limited APIs**: they either fail to involve real-world APIs (e.g., RESTAPI) or consider only a small scope of APIs with poor diversity; (2) **constrained scenario**: existing works are confined to instructions that only involve one single tool. Besides, they often assume that users manually specify the ideal API set for a given instruction in advance, which is infeasible in practice; In addition, some works do not even execute APIs to obtain real responses, which serve as important information for subsequent model planning.
>
> + **Our Unique Setting and Rich Resources**: Different from exsiting works, our ToolBench, stands out due to its comprehensive and diverse collection of **over 16,000** REST APIs from RapidAPI. For each API, we crawl detailed API documents from RapidAPI, including the functionality descriptions, required parameters, code snippets for API calls, etc. **This level of diversity in dataset construction is unprecedented in previous studies**. We believe the resource we provide represents a significant advancement in the field of tool learning research. Its diversity and comprehensiveness offer a robust platform for future studies focused on enhancing tool-use capabilities in LLMs.

---

> ### Author Response · Authors · 2023-11-15
> **Response - 2**
>
> A1-2: (continuing A1-1) Thirdly, contrary to the view that our approach lacks technical novelty, we have introduced several **innovative elements** in our methodology. Since we target a unique setting, our dataset construction process is also very different from previous works:
>
> + **Focus on Diversity and Multi-tool Usage**: Unlike previous studies, our work in ToolLLM places a strong emphasis on two critical aspects: **diversity** and **multi-tool usage**. We aim to train LLMs to handle a wide range of API usage scenarios, thereby enhancing their generalizability and robustness. Furthermore, by focusing on multi-tool usage, we mirror real-world situations that often require the interplay of multiple tools. This approach significantly improves the practical applicability and flexibility of LLMs in real-world scenarios.
>
> + **Innovative Instruction Crafting Methodology**: Our methodology for crafting instructions is notably different from prior works. Instead of brainstorming instructions from scratch and then searching for relevant APIs, we adopt a reverse engineering approach. We sample different combinations of APIs and craft various instructions that involve them. This strategy ensures **comprehensive coverage for all collected APIs** and leads to the generation of instructions that are more aligned with real-world application scenarios.
>
> + **Leveraging RapidAPI Hierarchy for Multi-tool Instructions**: To address the challenge of creating realistic multi-tool instructions, we use the RapidAPI hierarchy to identify tool relationships. This aids in generating multi-tool instructions that are not only diverse but also functionally coherent. Our approach to generating intra-category and intra-collection multi-tool instructions is a novel strategy that effectively tackles the sparsity issue often encountered in random sampling of tool combinations.
>
> In addition, we'd also like to highlight the unique and innovative aspects of the solution path annotation process, which we believe significantly contribute to the novelty of our work:
>
> + **Function Call Feature Utilization**: We innovatively leverage the function call feature of ChatGPT by treating each API as a special function and feeding its documentation into ChatGPT's function field. To the best of our knowledge, we are **the first** to introduce function calling feature into the filed of tool learning. This approach fully exploits ChatGPT’s capability of tool use.
>
> + **Depth First Search-based Decision Tree (DFSDT)**: Existing works adopted either CoT or ReACT for model reasoning, which cannot fully elicit the capabilities stored in LLMs and thus fail to handle complex instructions in tool learning. To overcome the limitations of existing methods like CoT and ReACT, we propose a novel Depth First Search-based Decision Tree. This method allows ChatGPT to assess different reasoning paths, choose promising directions, or abandon ineffective ones. Our DFSDT approach significantly enhances the exploration of the action space, increasing the likelihood of finding valid solution paths. In experiments, DFSDT significantly improves the annotation efficiency and successfully completes those complex instructions that cannot be fulfilled using ReACT.
>
> Lastly, to provide a comprehensive and objective assessment, we have conducted rigorous benchmarking against established models and released an innovative open-source model. Specifically:
>
> + **Benchmarking Against Established Models**: To assess the tool-use capabilities of LLMs, we develop an automatic evaluator, ToolEval. We demonstrate that ToolEval achieves a high correlation with human evaluation and provides a robust, scalable, and reliable assessment for machine tool use. Our comparative analysis using ToolEval offers a new perspective on how LLaMA, fine-tuned on ToolBench, measures up against well-established models (Claude, Davinci, ChatGPT, GPT-4). This benchmarking is essential for understanding the relative progress in the field and provides a practical reference point for future research.
>
> + **A Capable Open-source Model Released**: We provide a capable model ToolLLaMA, which demonstrates a compelling capability to handle both single-tool and complex multi-tool instructions. ToolLLaMA outperforms Text-Davinci-003 and Claude-2, achieves comparable performance to the ChatGPT, and is only slightly inferior to GPT4. Besides, ToolLLaMA exhibits robust generalization to previously unseen APIs.
>
> Considering these points, our work represents a significant and concurrent contribution to the field of instruction-tuning for tool learning. We believe that the concurrent development and unique aspects of our study demonstrate the novelty and research value of our work.

---

> ### Author Response · Authors · 2023-11-15
> **Response - 3**
>
> Q2: The author proposed "Depth First Search-based Decision Tree" (DFSDT) that seems to outperform ReAct prompting. However, the technique is closely related to self-consistency + CoT and Tree-of-thoughts.
>
> A2: Thank you for your insightful comments regarding our DFSDT method. Below are our responses to your concerns:
>
> + **Difference with Tree-of-thoughts (ToT)**: As mentioned in section 4, ToT is our concurrent work and there is a huge difference between ToT and DFSDT. Our DFSDT targets general decision-making problems where the decision space is infinite, compared to ToT's relatively simple tasks that can be addressed by brute-force search, such as Game of 24 and Crosswords. The distinct target between DFSDT and ToT determines the significant difference in the implementation details.
>
> + **Difference with self-consistency**: Self-consistency can be seen as performing multiple times of ReACT and performing a majority vote. This method is similar to one of our baseline in Table 3, i.e., ReACT@N. As mentioned in Section 3.1, ReACT@N conducts multiple times of ReACT until the total costs reach the same level of DFSDT. Once a valid solution is found by ReACT@N, we deem it a pass. Therefore, it resembles a form of self-consistency under our experimental setup. From Table 4, we can see that DFSDT outperforms this baseline by a large margin, which shows that our DFSDT performs much better than the self-consistency baseline. The reason for DFSDT's superiority lies in its ability to judge multiple reasoning paths during exploration, instead of after the exploration (i.e., the case of self-consistency). Hence DFSDT can detect and rectify errors earlier in the decision-making process, avoiding unnecessary exploration of mistaken path.
>
> + **Performance Comparison**: For performance comparison between DFSDT, ToT, and self-consistency, we borrow the results from a follow-up paper of our work [1]. They compared two versions of ToT (BFS and DFS) with our DFSDT. They implemented ToT the same as the original ToT paper. Below is the results on three different benchmarks (Game of 24, WebShop, and our ToolBench):
>
> | model                   | Game of 24 | WebShop | IToolBench |
> |-------------------------|----------|----------|---------|
> | CoT@3 (self-consistency)  | 7.00       | 56.45       | 31.20      |
> | ToT (BFS)  | 11.00       | 50.20       | 38.00      |
> | ToT (DFS) | 14.00       | 55.60       | 45.58      |
> | DFSDT (Ours)               | 29.00       | 57.25       | 50.20      |
>
> From the above results, we can see that our DFSDT outperforms the baseline of self-consistency and ToT in three benchmarks, demonstrating DFSDT’s superiority. Besides the performance, the authors of work [1] also show that our DFSDT is far more efficient than ToT and CoT within limited API calls (Figure 2 in work [1]), demonstrating DFSDT’s excellent efficiency in finding a good solution path. For more detailed discussion on the difference between these works, please kindly refer to paper [1].
>
> Besides elaborating on the difference between DFSDT and previous methods, we would also like to highlight the unique novelty of our DFSDT:
>
> + **Innovative Approach to Overcome Limitations of Existing Methods**: Our DFSDT was developed in response to identified limitations in CoT and ReACT methods, notably the issues of error propagation and limited exploration. Unlike these methods, which explore a single direction and can be trapped in faulty loops, DFSDT expands the search space, assessing multiple reasoning paths. This allows for a more comprehensive exploration of potential solutions, reducing the risk of error propagation and increasing the likelihood of finding valid solution paths.
>
> + **Explicit Encouragement for Diverse Node Generation**: During the node expansion process in our DFSDT, we prompt ChatGPT with information from previously generated nodes and explicitly encourage the generation of distinct nodes. This strategy ensures a diverse exploration of potential solutions, further expanding the search space.
>
> + **Flexibility and Versatility**: The design of our DFSDT allows it to degrade to ReACT for simpler instructions, making it as efficient as ReACT in these cases. For more complex instructions, it retains the full capabilities of a classical DFS search. This versatility ensures that our approach is applicable across a wide range of instruction complexities, enhancing the model's utility.
>
> In conclusion, our DFSDT presents significant advancements and distinct features. DFSDT is specifically designed to address complex, general decision-making problems with an infinite decision space, setting it apart from the more limited scopes of ToT and self-consistency approaches.
>
> [1] Anonymous: Rational Decision-Making Agent with Internalized Utility Judgment   https://openreview.net/forum?id=l1pNNQSzZv

---

> ### Author Response · Authors · 2023-11-15
> **Response - 4**
>
> Q3: It is also unclear how DFSDT will perform for other types of LLMs not in the category of LLaMA / GPT / Claude.
>
> A3: Thank you for raising a pertinent question regarding the performance of our DFSDT method across different types of LLMs beyond LLaMA, GPT, and Claude-2.
>
> In our study, we have meticulously conducted experiments with **seven** representative LLMs, encompassing a diverse range of structures and capabilities. These include models within the LLaMA architecture such as Alpaca, Vicuna, and our own ToolLLaMA, as well as models from the GPT series including Text-Davinci, GPT-3.5, and GPT-4, and the Claude-2 model. This selection of models was intentionally varied to provide a comprehensive evaluation of DFSDT's performance across different LLM frameworks.
>
> The results from these experiments clearly indicate that DFSDT is not only effective but also adaptable across different LLM architectures. This adaptability is crucial, as it demonstrates DFSDT's feasibility and potential for wide applicability in the field of LLMs. The success of DFSDT across this varied set of models underscores its robustness and the generalizability of its underlying principles.
>
> By demonstrating the efficacy of DFSDT across diverse LLM architectures, we affirm its potential as a versatile and reliable method for decision-making in LLM applications. This broad applicability is particularly important in the rapidly evolving landscape of LLMs, where the ability to adapt and perform across different model architectures is essential.
>
> To address your concern, we further test on another well-known LLM: ChatGLM.
>
> | model                   | pass rate | win rate |
> |-------------------------|----------|----------|
> | CoT  | 2       | 10       |
> | DFSDT  | 12       | 26       |
>
> From the above results, we again demonstrate that DFSDT surpasses CoT for ChatGLM. This reveals the superiority of our DFSDT on various LLM architectures.

---

> ### Author Response · Authors · 2023-11-15
> **Response - 5**
>
> Q4: The evaluation metrics are based on ChatGPT, which is highly unreliable and may favor models tuned with ChatGPT-prompted datasets. The author mentioned in Appendix A.5 (Paragraph "Comparing Human Evaluation and ToolEval") that Our ChatGPT evaluator demonstrates a high agreement of 87.1% in pass rate and 80.3% in win rate with human annotators. This shows that there is close to 20% disagreement between human evaluation and ChatGPT evaluation. This is a large discrepancy and should not be ignored.
>
> A4: Thank you for your observations regarding the use of ChatGPT-based evaluation metrics in our study. We understand your concerns about the reliability of these metrics and the noted discrepancy between our ChatGPT evaluator and human annotators. Below are our responses to this concern:
>
> + **ToolEval Achieves Comparable Performance to Humans**: According to our statistics, ChatGPT achieves a high agreement rate of 80.3% in win rate with human annotators, similar to the level among human experts (83.54% on average). This high level of agreement rate indicates that ChatGPT’s annotations **are in line with human judgment and are a reliable indicator of performance**. It is important to note that achieving complete alignment between human and automated evaluations in tool learning evaluation is challenging due to the subjective nature of certain judgments and the inherent variability in human evaluation. In addition, the fact that under our evaluation, GPT4 > GPT3.5 > Davinci (both pass rate and win rate) also reflects the reliability of our ToolEval.
>
> + **ChatGPT-based Evaluation as the Common Practice**: Employing ChatGPT for automatic evaluation is a common practice in the field of instruction-tuned LLMs [1,2,3,4]. This approach is not unique to our research but is **widely adopted** due to its effectiveness in providing quick and reasonably accurate assessments of model performance. The use of ChatGPT for evaluation in our research is driven by a need for efficiency and scalability in the evaluation process.
>
> + **We also Apply Evaluation other than ChatGPT-based Evaluation**: It should be noted that we do not rely solely on ChatGPT-based evaluation in this paper. For the out-of-distribution generalization experiments on APIBench, we use the **ground truth** to calculate the hallucination and ast accuracy. The promising results reflect the strong capabilities of our ToolLLaMA.
>
> In addition, when performing ChatGPT-based evaluation, we have performed a **rigorous analysis** of the evaluation ability of ChatGPT in our setting. Specifically, we have implemented specific strategies to mitigate some of these biases:
>
> + **Multi-Sample Strategy**: As mentioned in appendix A.5, to reduce individual biases, we employ a multi-sample strategy where multiple instances of evaluations (at least four) are aggregated. This approach ensures that the final output is not overly influenced by the peculiarities of a single model's response, providing a more balanced and representative result.
>
> + **Order-Switching Mechanism**: To counteract order bias, we systematically vary the order of options presented to the evaluator in different runs. This order-switching mechanism ensures that the sequence in which options are presented does not unduly influence the evaluation outcomes.
>
> In conclusion, while acknowledging the limitations of ChatGPT-based evaluation metrics, we believe that our approach provides a reliable, efficient, and scalable method for assessing LLM performance. The substantial agreement with human annotators underscores the validity of our findings, and we remain open to integrating more advanced evaluation techniques as they become available.
>
> [1] Liu, Yang, et al. "Gpteval: Nlg evaluation using gpt-4 with better human alignment." arXiv preprint arXiv:2303.16634 (2023).
>
> [2] Dubois, Yann, et al. "Alpacafarm: A simulation framework for methods that learn from human feedback." arXiv preprint arXiv:2305.14387 (2023).
>
> [3] Chiang, Wei-Lin, et al. "Vicuna: An open-source chatbot impressing gpt-4 with 90%* chatgpt quality." See https://vicuna. lmsys. org (accessed 14 April 2023) (2023).
>
> [4] Chiang, Cheng-Han, and Hung-yi Lee. "Can Large Language Models Be an Alternative to Human Evaluations?." arXiv preprint arXiv:2305.01937 (2023).

---

> ### Author Response · Authors · 2023-11-15
> **Response - 6**
>
> Q5: How does the models in Table 4 compare with each other if we adopt human evaluation?
>
> A5: Recognizing the importance of human judgment in validating model evaluations, we commit to including corresponding human evaluations in our revised paper. This will provide an additional layer of verification to our results, ensuring that they are not only efficient but also accurately reflect the true capabilities of the evaluated models. Due to the time limit of the rebuttal period, we have provided part of the results in the following:
>
> ## Win Rate: ChatGPT-DFSDT vs. ChatGPT-ReACT
> | Evaluation Method           | I1-instruction | I1-Tool | I1-Category | I2-Instruction | I2-Category | I3-Instruction | Average    |
> |-----------------------------|----------------|---------|-------------|----------------|-------------|----------------|------------|
> | Human Evaluation            | 55             | 58      | 61          | 75             | 61.5        | 69             | 63.3 |
> | ChatGPT Evaluation          | 60.5           | 62      | 57.3        | 72             | 64.8        | 69             | 64.3       |
>
> ## Pass Rate (Evaluate by ChatGPT)
> | Model           | I1-instruction | I1-Tool | I1-Category | I2-Instruction | I2-Category | I3-Instruction | Average |
> |-----------------|----------------|---------|-------------|----------------|-------------|----------------|---------|
> | ChatGPT-DFSDT   | 54.5           | 65      | 60.5        | 75             | 71.5        | 62             | 64.8    |
> | ToolLLaMA-DFSDT | 57             | 61      | 62          | 77             | 77          | 66             | 66.7    |
> | GPT4-DFSDT      | 60             | 71.5    | 67          | 79.5           | 77.5        | 71             | 71.1    |
>
> ## Pass Rate (Evaluate by Human)
> | Model           | I1-instruction | I1-Tool | I1-Category | I2-Instruction | I2-Category | I3-Instruction | Average      |
> |-----------------|----------------|---------|-------------|----------------|-------------|----------------|--------------|
> | ChatGPT-DFSDT   | 50             | 60      | 63          | 70             | 68          | 52             | 60.5         |
> | ToolLLaMA-DFSDT | 60             | 64      | 60          | 76             | 70          | 56             | 64.33333333  |
> | GPT4-DFSDT      | 64             | 66      | 64          | 76             | 76          | 62             | 68           |
>
> **Consistency Between Human and ChatGPT Evaluations**: Our results indicate a reasonable level of consistency between human evaluations and ChatGPT evaluations, particularly in terms of Win Rate. This consistency lends credibility to our automated evaluation methods while also providing the nuanced understanding that only human judgment can offer.
>
> We promise to include these results in the camera-ready version. The inclusion of human evaluation in our revised paper emphasizes our commitment to a comprehensive assessment of model capabilities. By combining automated and human evaluations, we can provide a more balanced and thorough understanding of each model's strengths and weaknesses. Thanks again for your suggestive comments!

---

> ### Author Response · Authors · 2023-11-20
> **Follow-Up: Seeking Further Feedback**
>
> Dear Reviewer, I hope you're doing well. Following up on our recent exchange regarding this paper, I wanted to check if there are any further concerns or feedback from your side. Your insights are invaluable to us, and we're keen to address any remaining issues.

---

### Official Review · Reviewer_dyqS · 2023-11-02

**Soundness:** 3 good
**Presentation:** 2 fair
**Contribution:** 3 good
**Rating:** 8
**Confidence:** 4

**Summary:**

This paper introduces ToolLLM, a whole pipeline for creating and evaluating instruction-tuned language models that can use tools.

The authors created ToolBench, an instruction-tuning dataset by i) collecting a large number of real-world APIs from multiple categories; ii) creating instructions with seed demonstrations and tool sets; iii) annotate the api call solution paths with ChatGPT.

They fine-tuned several baselines on ToolBench and evaluated them with the proposed ToolEval pipeline, which also uses ChatGPT to evaluate whether the solution path is successful and whether the solution path is better than the one annotated by ChatGPT.

Additionally, they also propose to use an API retriever and a tree-search algorithm (DFSDT) during inference.

**Strengths:**

1. The scale of ToolBench is unprecedented compared to previous tool learning datasets. It has more APIs, more tools, more task instances. This makes ToolBench much closer to real-world settings of tool-augmented language models.
2. The authors conducted extensive experiments to show that their instruction-tuning pipeline enables language models to generalize to new tools and new domains without seeing them in the fine-tuning dataset.

**Weaknesses:**

### 1. The current annotation/evaluation pipeline is not rigorous and may lead to false impressions about model’s performance.

ToolBench is entirely annotated by ChatGPT. I understand that the annotations of the instruction-tuning data don’t have to be perfect, but I think as a benchmark for evaluation, it should be held to higher standards.

ToolEval solely relies on ChatGPT to evaluate whether a solution path “passes” and whether it “wins” the solution path from ChatGPT.

While I appreciate the authors’ efforts in comparing ChatGPT’s evaluation with human subjects’, I think the current evaluation pipeline is problematic because of the following reasons:

- Many tool-related tasks can have a definite set of correct answers or correct solution paths. For these tasks, the real correct answers should be used to judge the correctness of generated answers. Relying on ChatGPT’s annotation of win rates and pass rates can lead to over-confidence about wrong answers that look satisfactory.
- ChatGPT (and other language models) as an evaluator is known to have order bias (gives preference to an option based on their order), egocentric bias (prefers its own outputs), length bias [1], selection bias [2]. It also chooses style over substance [3]. I would expect more discussion on if and how ToolEval mitigates these biases.
- The evaluation rules for pass rates and win rates seem too complex even for human annotators to follow, which severely undermines my confidence in your human evaluation results. According to Appendix A.5, the rules to determine whether a solution path gets a “pass” form a 3-layer decision tree with as many as 10 leaves and each decision in the tree requires some non-trivial and thorough examination of the instructions, the available APIs and the solution path. I would really love to learn more about your human evaluation process. For example, did the annotators only submit the final flag of pass/fail/unsure or did they also submit all the decisions they made to get to the final results? Is there any evidence that can show that human annotators were faithfully following your rules instead of relying on human cognitive biases? If the human annotations themselves were not reliable, nor would the correlation between human and ChatGPT be good enough.
- Win rates are computed by comparing model generation with annotations from ChatGPT+ReAct. Could inference algorithms that are more similar to the annotation pipeline (instead of better) lead to better results?

**Reference**

[1] Koo, Ryan, et al. "Benchmarking Cognitive Biases in Large Language Models as Evaluators." *arXiv preprint arXiv:2309.17012* (2023).

[2] Zheng, Chujie, et al. "On Large Language Models' Selection Bias in Multi-Choice Questions." *arXiv preprint arXiv:2309.03882* (2023).

[3] Wu, Minghao, and Alham Fikri Aji. "Style over substance: Evaluation biases for large language models." *arXiv preprint arXiv:2307.03025* (2023).

### 2. The evaluation results are constantly changing over time, making it very expensive and difficult to compare new methods and older baselines. This also ruins the reproducibility of the evaluation pipeline.

I appreciate the authors’ consideration about the temporal variability on RapidAI.

> *Considering the API’s temporal variability on RapidAPI and the infinite potential solution*
*paths for an instruction, it is infeasible to annotate a fixed ground-truth solution path for each test instruction. Moreover, when comparing different models, it is crucial to ensure they employ the same version of APIs during evaluation.*
>

I think this is also part of the reason why they used ChatGPT as an evaluator instead of using ground truth annotation.

However, I still think it’s problematic.

First, people are not able to know roughly how good a method is by looking at the reported pass rates and win rates, because the reported results can only be compared with the results evaluated during the same time period.

Second, it makes evaluation much more difficult. As the authors pointed out, each evaluation run needs to use the same version of APIs to ensure fair comparison. This means every new method that needs evaluation on ToolEval must also run every baseline they compare to in a very short period of time. If some highly variable API were used (for example, APIs that query the availability of restaurants or realtime weather), this period could be as short as a few hours. This creates an extremely heavy burden for developers and researchers, because not only do they need to run all evaluation experiments multiple times, they also have to run them in parallel.

Therefore, at the very least, I would expect some qualitative analysis of the tool set that can point out how many task instances involve these temporally variable tools and how much temporal variability impacts the evaluation results over time.

I think a better solution to this problem might be creating a “snapshot” of the API call results at a certain time and release this snapshot with the evaluation suite. This way, it’s much easier to get reproducible results.

### 3. The comparison between DFSDT and ReAct.

Why isn’t DFSDT always better than ReAct? According to your description, it seems that ReAct is a special case of DFSDT where the branching factor is 1. Therefore, I expected DFSDT to be better than ReAct in most cases. In other words, the win rate for ChatGPT+DFSDT against ChatGPT+ReAct should be close to 100%. According to the first two rows in Table 4, that is clearly not the case. Is there some explanation on why that happened?

**Questions:**

How were ToolBench split into tuning and evaluation subsets? Could you explain more about this part?

---

> ### Author Response · Authors · 2023-11-14
> **Response - 1**
>
> Q1: Many tool-related tasks can have a definite set of correct answers or correct solution paths. For these tasks, the real correct answers should be used to judge the correctness of generated answers.
>
> A1: Most of our instructions (tool-related tasks) **do not** have a definite set of correct answers or solution paths. We conducted human evaluation on 1000 instructions, and found that **86.7%** of them do not have a definite ground truth answer or solution path. The reasons are listed in the following:
>
> + **Temporal Variability of API Responses**: Many of our tool-related tasks are designed to interact with real-world data, which is inherently dynamic. For instance, an instruction about today's weather will yield different answers at different times, reflecting the ever-changing nature of weather conditions. **This temporal variability is a fundamental characteristic of the real-world APIs we are evaluating, making it impractical to define a single 'correct' answer or solution path for such tasks**.
>
> + **Infinite Solution Paths Due to API Diversity**: The vast array of available APIs (over 16,000) and their potential combinations contribute to **a multitude of valid solution paths for a given query**. Different APIs can interact in varied and sometimes unexpected ways, leading to multiple valid approaches to a given problem. **This complexity is reflective of real-world scenarios** where different tools and data sources are combined to solve complex tasks. As such, **our evaluation process is closely aligned with practical applications**, where there is often more than one way to achieve a desired outcome. Besides, in real scenarios, numerous APIs available offer similar functionalities or data (e.g., Bing translation and Google translation). This substitutability significantly increases the number of potential combinations and solution paths for a given task.
>
> + **Infinite Final Answers in Information Synthesis**: Even if a standard 'solution path' of API calls were to be established, the synthesis of the information could still vary. For example, in response to an instruction like "find the latest trends in technology," different APIs might offer varying perspectives—ranging from tech news to emerging products, to usage statistics. The LLM's role is to synthesize this diverse information into a coherent response that aligns with the user's query. This synthesis involves judgment and contextual understanding, which naturally leads to **multiple correct interpretations or presentations of the data**.
>
> In conclusion, while the idea of using a set of definite correct answers is appealing for simplicity, **it does not align with the reality of the tasks and APIs we are examining**. Our evaluation framework is designed to reflect real-world conditions, where answers are often not static but are influenced by temporal changes, the diversity of data sources, and the need for information synthesis. We believe that our approach offers a more accurate and practical assessment of LLM performance in real-world scenarios.

---

> ### Author Response · Authors · 2023-11-14
> **Response - 2**
>
> Q2: Relying on ChatGPT’s annotation of win rates and pass rates can lead to over-confidence about wrong answers that look satisfactory.
>
> A2: There is **no concrete evidence** that ChatGPT leads to over-confidence in our setting, although we appreciate your concern. Below are additional responses to this concern:
>
> + **ChatGPT-based Evaluation as the Common Practice**: Employing ChatGPT for automatic evaluation is a common practice in the field of instruction-tuned LLMs [1,2,3,4]. This approach is not unique to our research but is **widely adopted** due to its effectiveness in providing quick and reasonably accurate assessments of model performance. The use of ChatGPT for evaluation in our research is driven by a need for efficiency and scalability in the evaluation process.
>
> + **ToolEval Achieves Comparable Performance to Humans**: According to our statistics, ChatGPT achieves a high agreement rate of 80.3% in win rate with human annotators, similar to the level among human experts (83.54% on average). This high level of agreement rate indicates that **ChatGPT’s annotations are in line with human judgment** and are a reliable indicator of performance. The fact that under our evaluation, GPT4 > GPT3.5 > Davinci (both pass rate and win rate) also reflects the reliability of our ToolEval.
>
> + **Human Evaluation can also be Applied**: Recognizing the importance of human judgment in validating model evaluations, we commit to including corresponding human evaluations in our revised paper. This will provide an additional layer of verification to our results, ensuring that they are not only efficient but also accurately reflect the true capabilities of the evaluated models. Due to the time limit of the rebuttal period, **we promise to include these results in the camera-ready version**.
>
> Last but not least, it should be noted that **we do not rely solely on ChatGPT-based evaluation in this paper**. For the out-of-distribution generalization experiments on APIBench, we use the **ground truth** to calculate the hallucination and ast accuracy. The promising results reflect the strong capabilities of our ToolLLaMA.
>
> [1] Liu, Yang, et al. "Gpteval: Nlg evaluation using gpt-4 with better human alignment." arXiv preprint arXiv:2303.16634 (2023).
>
> [2] Dubois, Yann, et al. "Alpacafarm: A simulation framework for methods that learn from human feedback." arXiv preprint arXiv:2305.14387 (2023).
>
> [3] Chiang, Wei-Lin, et al. "Vicuna: An open-source chatbot impressing gpt-4 with 90%* chatgpt quality." See https://vicuna. lmsys. org (accessed 14 April 2023) (2023).
>
> [4] Chiang, Cheng-Han, and Hung-yi Lee. "Can Large Language Models Be an Alternative to Human Evaluations?." arXiv preprint arXiv:2305.01937 (2023).

---

> ### Author Response · Authors · 2023-11-14
> **Response - 3**
>
> Q3: ChatGPT as an evaluator is known to have order bias (gives preference to an option based on their order), egocentric bias (prefers its own outputs), length bias [1], selection bias [2]. It also chooses style over substance [3]. I would expect more discussion on if and how ToolEval mitigates these biases.
>
> A3: Continuing A2, we are aware of the biases inherent in LLMs. We have performed a **rigorous analysis** of the evaluation ability of ChatGPT in our setting. Specifically, we have implemented specific strategies to mitigate some of these biases:
>
> + **Multi-Sample Strategy**: As mentioned in appendix A.5, to reduce individual biases, we employ a multi-sample strategy where multiple instances of evaluations (at least four) are aggregated. This approach ensures that the final output is not overly influenced by the peculiarities of a single model's response, providing a more balanced and representative result.
>
> + **Order-Switching Mechanism**: To counteract order bias, we systematically vary the order of options presented to the evaluator in different runs. This order-switching mechanism ensures that the sequence in which options are presented does not unduly influence the evaluation outcomes.
>
> We recognize that mitigating biases such as egocentric and selection bias is challenging and requires ongoing effort. We plan to conduct further analysis and implement additional measures to address these biases in future iterations of our research. The complexities of these biases and our commitment to addressing them will be discussed more extensively in our revised paper.
>
> Last but not least, it's noteworthy that our approach, in terms of rigorously addressing potential biases and implementing a multifaceted evaluation strategy, **goes beyond what is commonly undertaken in similar studies**. We understand and appreciate the high standards set by the reviewer, as he/she reflects the advancing nature of this field. However, we also believe it's important to recognize that **our efforts to mitigate biases and validate our evaluation methods are already quite comprehensive compared to many existing works**. We agree that there is always room for further improvement and refinement, and we are committed to pursuing these in our ongoing research. At the same time, we believe that our current approach (multi-sample strategy and order-switching mechanism) represents a meaningful advancement in addressing the challenges inherent in evaluating LLMs.

---

> ### Author Response · Authors · 2023-11-14
> **Response - 4**
>
> Q4: The evaluation rules for pass rates and win rates seem too complex even for human annotators to follow, which severely undermines my confidence in your human evaluation results.
>
> A4: Thank you for elaborating on our human evaluation process, which is indeed a critical component of our research. We recognize your concern regarding the complexity of the decision tree used in the evaluation and its potential impact on the reliability of human annotations. We assure you that our evaluation process is designed to be **both comprehensive and rigorous**, with multiple safeguards to ensure the reliability and accuracy of human annotations.
>
> + **Detailed Annotator Training**: Our human annotators underwent comprehensive training to familiarize themselves with the detailed 3-layer decision tree used in our evaluation process. This training included multiple (**> 500**) example cases and interactive sessions to ensure a deep understanding of the criteria and the rationale behind each decision point. The training was designed to minimize subjective interpretations and ensure that each annotator applied the rules consistently. We also conduct **pre-annotation tests**, qualifying only those who achieve a high accuracy for actual annotation tasks. The final annotation team is composed of **6 NLP PhD students**.
>
> + **Submission of Decision-Making Process**: To ensure transparency and accountability, during the pre-annotation tests, annotators were required not only to submit the final flag of pass/fail/unsure **but also to document the decisions they made at each step of the decision tree**. This documentation allowed us to track and verify the rationale behind each annotation, ensuring that the rules were followed accurately.
>
> + **Ensuring Evaluators’ Adherence to Rules**: To ensure that evaluators were strictly adhering to the established rules, we conducted **periodic audits of the annotations**. These audits involved cross-checking a subset of evaluations against the decision tree to verify compliance with the established criteria. We also held regular review meetings where evaluators discussed challenging cases, ensuring a common understanding and application of the rules.
>
> + **Quality Checks**: We conducted regular quality checks on the annotations. These checks involved randomly selecting a subset of annotated solution paths and having them re-evaluated by a separate group of trained annotators.
>
> + **Annotation Interface**: To help annotators better conduct the evaluation, we also built a python-based annotation interface for them. This interface significantly eases their annotation process and improve the productivity.

---

> ### Author Response · Authors · 2023-11-14
> **Response - 5**
>
> Q5: Win rates are computed by comparing model generation with annotations from ChatGPT+ReAct. Could inference algorithms that are more similar to the annotation pipeline (instead of better) lead to better results?
>
> A5: Thanks for pointing it out! **Our evaluation method does not rely on any specific inference algorithm**, such as ChatGPT+ReACT, as a ground truth benchmark. Instead, we employ a pair-wise comparison approach, where we assess the relative merits and drawbacks of two different solution paths. **This pair-wise comparison approach is fundamentally independent of the choice of inference algorithm**. Whether we use ChatGPT+ReACT, DFSDT, or any other algorithm, the essence of our evaluation remains a relative comparison between solution paths. This ensures that our evaluation results are not biased towards the characteristics or performance of any specific inference algorithm.
>
> Actually, we have conducted more evaluation based on different inference algorithms, e.g. ChatGPT-DFSDT, and the results are shown in the following:
>
>
> | model                   | I1-Inst. | I1-Tool. | I1-Cat. | I2-Inst. | I2-Cat. | I3-Inst. | Average |
> |-------------------------|----------|----------|---------|----------|---------|----------|---------|
> | Text-Davinci-003-DFSDT  | 38       | 34       | 43      | 25       | 20      | 28       | 31.3    |
> | ToolLLaMA-API Retriever | 51       | 39       | 44      | 49       | 49      | 55       | 47.8    |
> | ToolLLaMA               | 43       | 42       | 46      | 55       | 46      | 50       | 47.0    |
>
> Using ChatGPT-DFSDT as the compared inference algorithm, we find that the win rate of ChatGPT-DFSDT > ToolLLaMA-DFSDT + API Retriever > ToolLLaMA-DFSDT > Davinci-DFSDT. Such results are aligned with the relative performance order presented in Table 4. It can be seen from the above that **the evaluation results of different inference algorithms are well-aligned**. These results also reflect that our ChatGPT-based evaluation is reliable.
>
> ----------------
>
> Q6: The evaluation results are constantly changing over time, making it very expensive and difficult to compare new methods and older baselines. This also ruins the reproducibility of the evaluation pipeline. First, people are not able to know roughly how good a method is by looking at the reported pass rates and win rates, because the reported results can only be compared with the results evaluated during the same time period.
>
> A6: We appreciate your concern regarding the temporal variability of evaluation results and its potential impact on reproducibility and comparability. We understand that the dynamic nature of APIs can lead to fluctuations in pass rates and win rates over time. However, we assert that **this variability does not undermine the reproducibility or relevance of our evaluation pipeline**. Our approach is designed to provide a realistic assessment of model performance in real-world scenarios, where such variability is inherent:
>
> + **Real-World Relevance**: Our evaluation framework is tailored to reflect real-world conditions, where APIs and data sources are inherently dynamic. The varying nature of APIs is a critical aspect of real-world applications, and our evaluation setup aims to capture this dynamism. Thus, the changing evaluation results offer a more accurate reflection of how models would perform in practical settings.
>
> + **Framework for Comparative Analysis**: Followers of this paper **do not need to refer to the specific numbers reported in this paper**. Instead, we encourage researchers and developers to use our released codes and evaluator to conduct their evaluations in the same temporal context as their models. This ensures that comparisons are made under similar conditions, providing a fair and relevant benchmarking of model performances. **The focus should be on the relative performance of different methods rather than the absolute numbers**, which are expected to vary over time.
>
> + **Guidance for Future Research**: In our revised paper, we will provide detailed guidelines on how to use our evaluation framework effectively. This will include instructions on how to compare new methods with existing baselines, such as ToolLLaMA, by conducting simultaneous evaluations to obtain comparable pass rates and win rates. This approach allows for a meaningful comparison of different methods, even as specific numbers change over time.
>
> + **Adding Contextual Information**: To further aid in reproducibility and comparability, we plan to include contextual information about the API versions and the temporal conditions under which the evaluations were conducted. This will provide future researchers with a reference point for understanding the conditions of our initial evaluations and for making informed comparisons with their own results.

---

> ### Author Response · Authors · 2023-11-14
> **Response - 6**
>
> Q7-1: Second, it makes evaluation much more difficult. Each evaluation run needs to use the same version of APIs to ensure fair comparison. This means every new method that needs evaluation on ToolEval must also run every baseline they compare to in a very short period of time. If some highly variable API were used (for example, APIs that query the availability of restaurants or realtime weather), this period could be as short as a few hours. This creates an extremely heavy burden for developers and researchers, because not only do they need to run all evaluation experiments multiple times, they also have to run them in parallel.
>
> Q7-2: I would expect some qualitative analysis of the tool set that can point out how many task instances involve these temporally variable tools and how much temporal variability impacts the evaluation results over time.
>
> A7: We appreciate your highlighting the challenges posed by the temporal variability of APIs in the evaluation process. To mitigate these challenges and assist researchers and developers, we have implemented the following solutions:
>
> + **RapidAPI Server Support**: Recognizing the difficulties in accessing and using RapidAPI, we **have established a dedicated RapidAPI server** on GitHub (due to anonymous issues, we cannot directly provide the link, but it can be easily found on the web). This server is designed to facilitate researchers in conducting their studies more efficiently. It **supports parallel API calls**, reducing the time and effort needed to run evaluations in tandem. Our commitment to maintaining and updating this server is unwavering, and we have collaborated with official RapidAPI staff to ensure its reliability and effectiveness. Up to now, we have helped **920 independent researchers** with their research, and our server has supported roughly **8600000 API requests** from these researchers. The response from the research community has been overwhelmingly positive, as indicated by **over 3700 stars** on our GitHub repository. We are committed to continuing this support. Our engagement with the community and ongoing improvements to the server demonstrates our dedication to facilitating research in this area. Promisingly, **we have seen some follow-up works to use our ToolEval as evaluation**, such as [1].
>
>
> + **Rate of API Updates**: We acknowledge the importance of your request for a detailed analysis of the impact of temporal variability on our API-based task instances. To provide clarity on this aspect, we have conducted a thorough analysis of the APIs used in our dataset by analyzing a total of 5979 API calls involving 1100 distinct instructions. Our analysis indicates that, on average, **only about 1.4 APIs from our dataset are updated every 24 hours**. This translates to approximately **0.7%** of the total APIs requiring updates each month. This relatively **low rate of change** suggests that the majority of the APIs we are using exhibit stable behavior over short-to-medium time periods.
>
> + **Monitoring and Updating Protocol**: we regularly monitor the APIs for significant changes. In cases where an API update might materially affect the task outcomes, we update our dataset and evaluation metrics accordingly. This proactive approach ensures that our evaluations remain relevant and accurate, even in the face of evolving API landscapes. Since we released our RapidAPI server, we have updated the backend system (including API documents, codes, etc.) for **4 times** within the past 2 months. We recognize that temporal variability is an inherent aspect of working with real-world APIs and data sources. We are committed to continuously improving our methods for tracking and accounting for these changes.
>
> In conclusion, while the temporal variability of APIs presents a unique challenge, we have taken significant steps to alleviate the burden on researchers and developers. Our RapidAPI server offers practical solutions for conducting fair and consistent evaluations. We will continue to refine these tools and explore new ways to support the research community in this rapidly evolving field.
>
> [1] Zhang, Kexun, et al. "Syntax Error-Free and Generalizable Tool Use for LLMs via Finite-State Decoding." arXiv preprint arXiv:2310.07075 (2023).

---

> ### Author Response · Authors · 2023-11-14
> **Response - 7**
>
> Q8: I think a better solution to this problem might be creating a “snapshot” of the API call results at a certain time and release this snapshot with the evaluation suite. This way, it’s much easier to get reproducible results.
>
> A8: We appreciate your suggestion to create a “snapshot” of the API call results to enhance reproducibility. However, due to the sheer volume and variability of API calls, **caching them effectively for future use is not feasible**. Each change in the parameters of an API call would render the cached data obsolete, making it challenging to create a comprehensive and enduring snapshot. For instance, an instruction like “what’s the weather today?” yields different results each day. Capturing a snapshot of such API calls on a specific day would only be representative of one day. They would quickly become outdated if the input parameter of “date” changes. This limitation reduces the utility of snapshots for evaluating methods that rely on real-time or frequently updated data.
>
> Furthermore, as previously discussed, the vast array of available APIs and their potential combinations create an almost infinite array of solution paths. Each task could be approached in numerous ways, with each unique combination of API calls yielding different results. This immense diversity in potential solution paths means that constructing a comprehensive snapshot that accounts for all possible scenarios is impractical.
>
> However, we recognize the importance of reproducibility in research and we suggest an alternative solution:
>
> + **Building a Simulated APIServer**: we can simulate the API responses using ChatGPT. Since we have obtained real API responses from almost all of our APIs, we can use these responses as in-context examples to generate a simulated API call by prompting. This can be easily deployed locally, simulating API interactions in a consistent manner. Researchers and developers can use this server to conduct evaluations without the complexities of real-time API variability. This approach allows for consistent, reproducible results, while still offering the flexibility to test against a range of simulated scenarios.
>
> We believe this alternative strikes a balance between the need for real-time data in evaluations and the necessity for reproducible research environments. It offers a practical solution to the challenges posed by the dynamic nature of APIs, while also aligning with the goals of fair and consistent evaluation across different research methodologies. This new server will be released soon.
>
> ---------
>
> Q9:  Why isn’t DFSDT always better than ReAct? According to your description, it seems that ReAct is a special case of DFSDT where the branching factor is 1. Therefore, I expected DFSDT to be better than ReAct in most cases. In other words, the win rate for ChatGPT+DFSDT against ChatGPT+ReAct should be close to 100%. According to the first two rows in Table 4, that is clearly not the case. Is there some explanation on why that happened?
>
> A9: We appreciate your question regarding the performance comparison between DFSDT and ReAct, and why DFSDT does not consistently outperform ReAct as might be expected. There are several factors that contribute to this observation:
>
> + **Inherent Randomness in LLMs**: One of the primary factors is the inherent randomness in large language models (LLMs), such as the temperature setting in generation. This randomness means that the first branch generated by DFSDT can vary significantly across different instances. As we have indicated in our paper, multiple iterations of ReAct can sometimes yield better results compared to a single iteration of DFSDT. This variation is due to the stochastic nature of LLM outputs, which can affect the initial branching and subsequent solution path development in DFSDT.
>
> + **LLM as Imperfect Judges**: The second factor is the reliance of DFSDT on the LLM to judge and select the best solution path. While DFSDT theoretically expands the search space for potential solutions, it ultimately depends on the LLM's judgment to select the most promising path. LLMs, despite their advanced capabilities, are not infallible judges. There are cases where the LLM might fail to identify the optimal solution path, leading to a scenario where DFSDT does not outperform ReAct.
>
> For more detailed analysis, discussion, and experiments on the performance of different searching algorithms (e.g., DFSDT, ToT, ReACT) and their difference, we refer you to one follow-up work of our paper: [1].
>
> [1] Ye, Yining, et al. "Large language model as autonomous decision maker." arXiv preprint arXiv:2308.12519 (2023).

---

> ### Author Response · Authors · 2023-11-14
> **Response - 8**
>
> Q11: How was ToolBench split into tuning and evaluation subsets? Could you explain more about this part?
>
> A11: Initially, we divide categories into training and testing categories. Then within the training category, tools are divided into training tools and testing tools, and within the training tools, APIs are classified as training APIs and testing APIs. Subsequently, instructions involving only training APIs are used to construct the training set. For constructing the testing set, as exemplified by the I1-instruction test set, to test the model's command generalization ability, the test data involves training APIs that are within the training tools of the training category. The I1-Tool involves APIs under testing tools in the training category, and the I1-Category involves APIs from tools within the testing category. The same logic applies to I2-Category, I2-Instruction, and I3-Instruction.
>
> **Last but not least**, we would like to express our gratitude for your comprehensive review and insightful comments on this paper. We sincerely believe that these suggestions will guide us better in the future research. However, **it should be noted that this paper is not merely an evaluation paper**, and we sincerely hope reviewers could perform a comprehensive evaluation of the contribution of this paper, many thanks!!
>
> The contribution of this paper is as follows:
>
> + We provide a comprehensive tool learning benchmark consisting of massive APIs. The benchmark supports both single-tool and multi-tool settings.
>
> + We gather 16,464 representational state transfer (REST) APIs from RapidAPI. For each API, we crawl detailed API documents from RapidAPI, including the functionality descriptions, required parameters, code snippets for API calls, etc.
>
> + We develop a novel DFSDT algorithm to bolster the planning and reasoning ability of LLMs. Compared with conventional ReACT, DFSDT enables LLMs to evaluate a multitude of reasoning paths and make deliberate decisions to either retract steps or proceed along a promising path. In experiments, DFSDT significantly improves the annotation efficiency and successfully completes those complex instructions that cannot be fulfilled using ReACT.
>
> + To assess the tool-use capabilities of LLMs, we develop an automatic evaluator, ToolEval. We demonstrate that ToolEval achieves a high correlation with human evaluation and provides a robust, scalable, and reliable assessment for machine tool use.
>
> + We provide a capable model ToolLLaMA, which demonstrates a compelling capability to handle both single-tool and complex multi-tool instructions. ToolLLaMA outperforms Text-Davinci-003 and Claude-2, achieves comparable performance to the ChatGPT, and is only slightly inferior to GPT4. Besides, exhibits robust generalization to previously unseen APIs.

---

> ### Author Response · Authors · 2023-11-16
> **Response - 9**
>
> Dear Reviewer,
>
> Recognizing the importance of human judgment in validating model evaluations, we commit to including corresponding human evaluations in our revised paper. This will provide an additional layer of verification to our results, ensuring that they are not only efficient but also accurately reflect the true capabilities of the evaluated models. Due to the time limit of the rebuttal period, we have provided part of the results in the following:
>
> ## Win Rate: ChatGPT-DFSDT vs. ChatGPT-ReACT
> | Evaluation Method           | I1-instruction | I1-Tool | I1-Category | I2-Instruction | I2-Category | I3-Instruction | Average    |
> |-----------------------------|----------------|---------|-------------|----------------|-------------|----------------|------------|
> | Human Evaluation            | 55             | 58      | 61          | 75             | 61.5        | 69             | 63.3 |
> | ChatGPT Evaluation          | 60.5           | 62      | 57.3        | 72             | 64.8        | 69             | 64.3       |
>
> ## Pass Rate (Evaluate by ChatGPT)
> | Model           | I1-instruction | I1-Tool | I1-Category | I2-Instruction | I2-Category | I3-Instruction | Average |
> |-----------------|----------------|---------|-------------|----------------|-------------|----------------|---------|
> | ChatGPT-DFSDT   | 54.5           | 65      | 60.5        | 75             | 71.5        | 62             | 64.8    |
> | ToolLLaMA-DFSDT | 57             | 61      | 62          | 77             | 77          | 66             | 66.7    |
> | GPT4-DFSDT      | 60             | 71.5    | 67          | 79.5           | 77.5        | 71             | 71.1    |
>
> ## Pass Rate (Evaluate by Human)
> | Model           | I1-instruction | I1-Tool | I1-Category | I2-Instruction | I2-Category | I3-Instruction | Average      |
> |-----------------|----------------|---------|-------------|----------------|-------------|----------------|--------------|
> | ChatGPT-DFSDT   | 50             | 60      | 63          | 70             | 68          | 52             | 60.5         |
> | ToolLLaMA-DFSDT | 60             | 64      | 60          | 76             | 70          | 56             | 64.33333333  |
> | GPT4-DFSDT      | 64             | 66      | 64          | 76             | 76          | 62             | 68           |
>
> **Consistency Between Human and ChatGPT Evaluations**: Our results indicate a reasonable level of consistency between human evaluations and ChatGPT evaluations, particularly in terms of Win Rate. This consistency lends credibility to our automated evaluation methods while also providing the nuanced understanding that only human judgment can offer.
>
> We promise to include these results in the camera-ready version. The inclusion of human evaluation in our revised paper emphasizes our commitment to a comprehensive assessment of model capabilities. By combining automated and human evaluations, we can provide a more balanced and thorough understanding of each model's strengths and weaknesses. Thanks again for your suggestive comments!

---

> ### Author Response · Authors · 2023-11-20
> **Follow-Up: Seeking Further Feedback**
>
> Dear Reviewer, I hope you're doing well. Following up on our recent exchange regarding this paper, I wanted to check if there are any further concerns or feedback from your side. Your insights are invaluable to us, and we're keen to address any remaining issues.

---

> > ### Comment · Reviewer_dyqS · 2023-11-20
> >
> > Thank you for your detailed response!
> >
> > It has addressed most of my concerns, and I do appreciate the huge efforts you put into this paper and the response. Therefore, I'm raising my score to 8.
> >
> > I would really appreciate it if you could put some parts of the response in the revision (you don't have to do it during the rebuttal period) and carry out what you suggested to improve the reproducibility (like building a simulated API server).

---

> > > ### Author Response · Authors · 2023-11-20
> > > **Response**
> > >
> > > Dear reviewer, thanks very much for your detailed comments. We will put most of our responses into our revised paper, and we will also release the simulated API server within two weeks (on GitHub). Thanks again for your valuable suggestions!

---

### Meta-Review · Area_Chair_drRW · 2023-12-06

**Metareview:**

The paper is clearly written and easy to understand. ToolBench's scale is unparalleled compared to prior datasets on tool usage. The authors have conducted thorough experiments, demonstrating that their instruction-tuning pipeline effectively enables language models to adapt to new tools and domains not seen in the fine-tuning dataset.

While there are ongoing concerns about the temporal variability of API responses and the differences between ChatGPT-based evaluation and human evaluation, ACs acknowledge the breadth and complexity of challenges in this field. Considering the benchmark's comprehensiveness and the effort to establish standardized metrics in tool usage, I recommend an accept (spotlight) rating.

**Justification For Why Not Higher Score:**

There are still concerns about the temporal variability of API responses and the differences between ChatGPT-based evaluation and human evaluation.

**Justification For Why Not Lower Score:**

ToolBench has been widely adopted in academics with high impact. The scale is unparalleled compared to prior datasets on tool usage.

---

### Decision · Program_Chairs · 2024-01-16

Accept (spotlight)